# Erwin: A Tree-based Hierarchical Transformer for Large-scale Physical Systems

Maksim Zhdanov [1]   Max Welling [1 2]   Jan-Willem van de Meent [1]

## Abstract

Large-scale physical systems defined on irregular grids pose significant scalability challenges for deep learning methods, especially in the presence of long-range interactions and multi-scale coupling. Traditional approaches that compute all pairwise interactions, such as attention, become computationally prohibitive as they scale quadratically with the number of nodes. We present Erwin, a hierarchical transformer inspired by methods from computational many-body physics, which combines the efficiency of tree-based algorithms with the expressivity of attention mechanisms. Erwin employs ball tree partitioning to organize computation, which enables linear-time attention by processing nodes in parallel within local neighborhoods of fixed size. Through progressive coarsening and refinement of the ball tree structure, complemented by a novel cross-ball interaction mechanism, it captures both fine-grained local details and global features. We demonstrate Erwin's effectiveness across multiple domains, including cosmology, molecular dynamics, PDE solving, and particle fluid dynamics, where it consistently outperforms baseline methods both in accuracy and computational efficiency.

## 1. Introduction

Scientific deep learning is tackling increasingly computationally intensive tasks, following the trajectory of computer vision and natural language processing. Applications range from molecular dynamics (MD) (Arts et al., 2023) and computational particle mechanics (Alkin et al., 2024b) to weather forecasting (Bodnar et al., 2024), where simulations often involve data defined on irregular grids with thousands to millions of nodes, depending on the required resolution and complexity of the system.

[1] AMLab, University of Amsterdam [2] CuspAI. Correspondence to: Maksim Zhdanov <m.zhdanov@uva.nl>.

*Proceedings of the 42nd International Conference on Machine Learning*, Vancouver, Canada. PMLR 267, 2025. Copyright 2025 by the author(s).

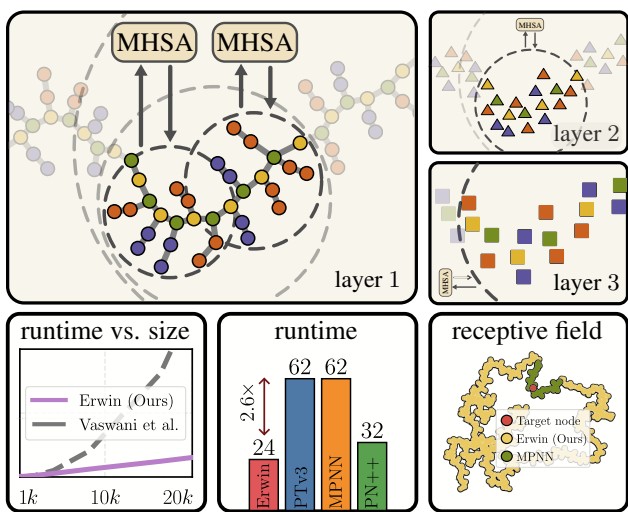

*Figure 1.* **Top:** Ball tree attention over a molecular graph. Multi-head self-attention (MHSA) is computed in parallel at fixed hierarchy levels (bold circles). In the following layers, the tree is progressively coarsened to learn global features, while the partition size is fixed. **Bottom:** Computational advantages of our model.

Such large-scale systems pose a significant challenge to existing methods that were developed and validated at smaller scales. For example, in computational chemistry, models are typically trained on molecules with tens of atoms (Kovács et al., 2023), while molecular dynamics simulations often exceed thousands of atoms. This scale disparity might result in prohibitive runtimes that render models inapplicable in high-throughput scenarios such as protein design (Watson et al., 2023) or screening (Fu et al., 2022).

A key challenge in scaling to larger system sizes is that computational methods that work well at small scales break down at larger scales. For small systems, all pairwise interactions can be computed explicitly, allowing deep learning models to focus on properties like equivariance (Cohen & Welling, 2016). However, this brute-force approach becomes intractable as the system size grows. At larger scales, approximations are required to efficiently capture both long-range effects from slowly decaying potentials and multi-scale coupling (Majumdar et al., 2020). As a result, models validated only on small systems often lack the architectural components necessary for efficient scaling.

This problem has been extensively studied in computational many-body physics (Hockney & Eastwood, 2021), where the need for evaluating long-range potentials for large-scale particle systems led to the development of sub-quadratic tree-based algorithms (Barnes & Hut, 1986; Carrier et al., 1988). These methods are based on the intuition that distant particles can be approximated through their mean-field effect rather than individual interactions (Pfalzner & Gibbon, 1996). The computation is then structured using hierarchical trees to efficiently organize operations at multiple scales. While highly popular for numerical simulations, these tree-based methods have seen limited adoption in deep learning due to poor synergy with GPU architectures.

Transformers (Vaswani et al., 2017), on the other hand, employ the highly optimized attention mechanism, which comes with the quadratic cost of computing all-to-all interactions. In this work, we combine the efficiency of hierarchical tree methods with the expressivity of attention to create a scalable architecture for processing large-scale particle systems. Our approach leverages ball trees to organize computation at multiple scales, enabling both local accuracy and global feature capture while maintaining *linear* complexity in the number of nodes.

The main contributions of the work are the following:

- We introduce ball tree partitioning for efficient point cloud processing, enabling linear-time self-attention through localized computation within balls at different hierarchical levels.
- We present Erwin, a hierarchical transformer that processes data through progressive coarsening and refinement of ball tree structures, effectively capturing both fine-grained local interactions and global features while maintaining computational efficiency.
- We validate Erwin's performance across multiple large-scale physical domains:
  - Capturing long-range interactions (cosmology)
  - Computational efficiency (molecular dynamics)
  - Expressivity on large-scale phenomena (PDE benchmarks, turbulent fluid dynamics)

achieving state-of-the-art performance in both computational efficiency and prediction accuracy.

## 2. Related Works: sub-quadratic attention

One way to avoid the quadratic cost of self-attention is to linearize attention by performing it on non-overlapping patches. For data on regular grids, like images, the Swin Transformer (Liu et al., 2021) achieves this by limiting attention to local windows with cross-window connections enabled by shifting the windows. However, for irregular data such as point clouds or non-uniform meshes, one first needs to induce a structure that will allow for patching. Several approaches (Liu et al., 2023; Sun et al., 2022) transform point clouds into sequences, most notably PointTransformer v3 (PTv3) (Wu et al., 2024b), which projects points into voxels and orders them using space-filling curves (e.g., Hilbert curve). While scalable, these curves introduce artificial discontinuities that can break local spatial relationships.

Particularly relevant to our work are hierarchical attention methods. In the context of 1D sequences, approaches like the H-transformer (Zhu & Soricut, 2021) and Fast Multipole Attention (Kang et al., 2023) approximate self-attention through multi-level decomposition: tokens interact at full resolution locally while distant interactions are computed using learned or fixed groupings at progressively coarser scales. For point clouds, OctFormer (Wang, 2023) converts spatial data into a sequence by traversing an octree, ensuring spatially adjacent points are consecutive in memory. While conceptually similar to our approach, OctFormer relies on computationally expensive octree convolutions, whereas our utilization of ball trees leads to significant efficiency gains.

Rather than using a hierarchical decomposition, another line of work proposes cluster attention (Janny et al., 2023; Alkin et al., 2024a; Wu et al., 2024a). These methods first group points into clusters and aggregate their features at the cluster centroids through message passing or cross-attention. After computing attention between the centroids, the updated features are then distributed back to the original points. While these approaches yield the quadratic cost only in the number of clusters, they introduce an information bottleneck at the clustering step that may sacrifice fine-grained details and fail to capture features at multiple scales - a limitation our hierarchical approach aims to overcome.

Beyond attention, several alternatives have been developed for processing large-scale systems with irregular geometries. Message-passing neural networks (Gilmer et al., 2017) typically address scalability through multi-level graph representations (Lam et al., 2023; Cao et al., 2023; Valencia et al., 2025). Additionally, recent works have explored sub-quadratic convolution-based architectures like Hyena (Moskalev et al., 2025) and state-space models (Zhang et al., 2025), which offer promising alternatives for efficient processing of geometric data without the computational overhead of attention mechanisms.

## 3. Background

Our work revolves around attention, which we aim to linearize by imposing structure onto point clouds using ball trees. We formally introduce both concepts in this section.

### 3.1. Attention

The standard self-attention mechanism is based on the scaled dot-product attention (Vaswani et al., 2017). Given a set $X$

of $N$ input feature vectors of dimension $C$, self-attention is computed as

$$\mathbf{Q}, \mathbf{K}, \mathbf{V} = \mathbf{X}\mathbf{W}_q, \mathbf{X}\mathbf{W}_k, \mathbf{X}\mathbf{W}_v$$

$$\text{Att}(\mathbf{Q}, \mathbf{K}, \mathbf{V}) = \text{softmax}\left(\frac{\mathbf{Q}\mathbf{K}^T}{\sqrt{C'}} + \mathcal{B}\right)\mathbf{V} \quad (1)$$

where $\mathbf{W}_q, \mathbf{W}_k, \mathbf{W}_v \in \mathbb{R}^{C \times C'}$ are learnable weights and $\mathcal{B} \in \mathbb{R}^{N \times N}$ is the bias term.

Multi-head self-attention (MHSA) improves expressivity by computing attention $H$ times with different weights and concatenating the output before the final projection:

$$\text{MHSA}(\mathbf{X}) = [\mathbf{Y}_1, \cdots, \mathbf{Y}_H]\mathbf{W}^O$$

$$\mathbf{Y}_i = \text{Att}(\mathbf{X}\mathbf{W}_q^i, \mathbf{X}\mathbf{W}_k^i, \mathbf{X}\mathbf{W}_v^i) \quad (2)$$

where $[\cdot, \cdots, \cdot]$ denotes concatenation along the feature dimension, and $\mathbf{W}_q^i, \mathbf{W}_k^i, \mathbf{W}_v^i \in \mathbb{R}^{C \times (C'/H)}$ and $\mathbf{W}^O \in \mathbb{R}^{C \times C'}$ are learnable weights.

The operator explicitly computes interactions between all elements in the input set without any locality constraints. This yields the quadratic computational cost w.r.t. the input set size $\mathcal{O}(N^2)$. Despite being heavily optimized (Dao, 2024), this remains a bottleneck for large-scale applications.

## 3.2. Ball tree

A ball tree is a hierarchical data structure that recursively partitions points into nested sets of equal size, where each set is represented by a ball that covers all the points in the set. Assume we operate on the $d$-dim. Euclidean space $\left(\mathbb{R}^d, ||\cdot||_2\right)$ where we have a point cloud (set) $P = \{\mathbf{p}_1, ..., \mathbf{p}_n\} \subset \mathbb{R}^d$.

**Definition 3.1** (Ball). A *ball* is a region bounded by a hypersphere in $\mathbb{R}^d$. Each ball is represented by the coordinates of its center $\mathbf{c} \in \mathbb{R}^d$ and radius $r \in \mathbb{R}_+$:

$$B = B(\mathbf{c}, r) = \{\mathbf{z} \in \mathbb{R}^d \mid ||\mathbf{z} - \mathbf{c}||_2 \le r\}. \quad (3)$$

We will omit the parameters $(\mathbf{c}, r)$ for brevity from now on.

**Definition 3.2** (Ball Tree). A *ball tree* $T$ on point set $P$ is a hierarchical sequence of partitions $\{L_0, L_1, ..., L_m\}$, where each level $L_i$ consists of disjoint balls that cover $P$. At the leaf level $i = 0$, the nodes are the original points:

$$L_0 = \{\{\mathbf{p}_j\} \mid \mathbf{p}_j \in P\}$$

For each subsequent level $i > 0$, each ball $B \in L_i$ is formed by merging two balls at the previous level $B_1, B_2 \in L_{i-1}$:

$$L_i = \{\{B_1 \cup B_2\} \mid B_1, B_2 \in L_{i-1}\} \quad (4)$$

such that its center is computed as the center of mass:

$$\mathbf{c}_B = \frac{|B_1|\mathbf{c}_1 + |B_2|\mathbf{c}_2}{|B_1| + |B_2|}$$

and its radius is determined by the furthest point it contains:

$$r_B = max\{||\mathbf{p} - \mathbf{c}_B||_2 \mid \mathbf{p} \in B_1 \cup B_2\}$$

where $|B|$ denotes the number of points contained in $B$.

To construct the ball tree, we recursively split the data points into two sets starting from $P$. In each recursive step, we find the dimension of the largest spread (i.e., the max − min value) and split at its median (Pedregosa et al., 2012), constructing covering balls per Def.3.2. For details, see Appendix Alg.1[1].

**Tree Completion**    To enable efficient implementation, we want to work with *perfect* binary trees, i.e., trees where all internal nodes have exactly two children and all leaf nodes appear at the same depth. To achieve this, we pad the leaf level of a ball tree with virtual nodes, yielding the total number of nodes $2^m$, where $m = \text{ceil}(\log_2(n))$.

### 3.2.1. Ball tree properties

In the context of our method, there are several properties of ball trees that enable efficient hierarchical partitioning:

**Proposition 3.3** (Ball Tree Properties). *The ball tree $T$ constructed as described satisfies the following properties:*

1. *The tree is a perfect binary tree.*

2. *At each level $i$, each ball contains exactly $2^i$ leaf nodes.*

3. *Balls at each level cover the point set*

$$\bigcup_{B \in L_i} B = P \quad \forall i \in \{0, ..., m\}.$$

**Proposition 3.4** (Contiguous Storage). *For a ball tree $T = \{L_0, L_1, ..., L_m\}$ on point cloud $P = \{\mathbf{p}_1, ..., \mathbf{p}_n\}$, there exists a bijective mapping $\pi : \{1, ..., n\} \to \{1, ..., n\}$ such that points belonging to the same ball $B \in L_i$ have contiguous indices under $\pi$.*

As a corollary, the hierarchical structure at each level can be represented by nested intervals of contiguous indices:

**Example.** *Let $P = \{\mathbf{p}_1, ..., \mathbf{p}_8\}$, then a ball tree $T = \{L_0, L_1, L_2, L_3\}$ is stored after the permutation $\pi$ as*

$L_3$

$L_2$

$L_1$

$L_0 = \pi(P) \quad \mathbf{p}_a \ \mathbf{p}_b \ \mathbf{p}_c \ \mathbf{p}_d \ \mathbf{p}_e \ \mathbf{p}_f \ \mathbf{p}_g \ \mathbf{p}_h$

---

[1]Note that since we split along coordinate axes, the resulting structure depends on the orientation of the input data and thus breaks rotation invariance. We will rely on this property in Section 4.1 to implement cross-ball connections.

The contiguous storage property, combined with the fixed size of balls at each level, enables efficient implementation through tensor operations. Specifically, accessing any ball $B \in L_i$ simply requires selecting a contiguous sequence of $2^i$ indices. For instance, in the example above, for $i = 2$, we select a:d and e:h to access the balls. Since the balls are equal in size, we can simply reshape $L_0$ to access any level. This representation makes it particularly efficient to implement our framework's core operations - ball attention and coarsening/refinement - which we will introduce next.

Another important property of ball trees is that while they cover the whole point set, they are not required to partition the entire space. Coupled with completeness, it means that at each tree level, the nodes are essentially associated with the same scale. This contrasts with other structures such as octrees that cover the entire space and whose nodes at the same level can be associated with regions of different sizes:

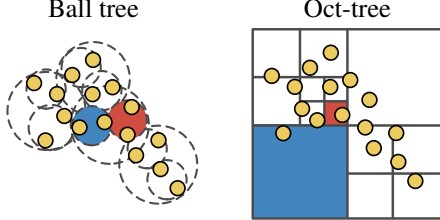

*Figure 2.* Ball tree vs. octree construction. Colors highlight the difference in scales for nodes including the same number of points.

## 4. Erwin Transformer

Following the notation from the Background Section 3.2, we consider a point cloud $P = \{\mathbf{p}_1, ..., \mathbf{p}_n\} \subset \mathbb{R}^d$. Additionally, each point is now endowed with a feature vector yielding a feature set $X = \{\mathbf{x}_1, ..., \mathbf{x}_n\} \subset \mathbb{R}^C$.

On top of the point cloud, we build a ball tree $T = \{L_0, ..., L_m\}$. We initialize $L_{\text{leaf}} := L_0$ to denote the current finest level of the tree. As each leaf node contains a single point, it inherits its feature vector:

$$X_{\text{leaf}} = \{\mathbf{x}_B = \mathbf{x}_i \mid B = \{\mathbf{p}_i\} \in L_{\text{leaf}}\} \quad (5)$$

### 4.1. Ball tree attention

**Ball attention** For each ball attention operator, we specify a level $k$ of the ball tree where each ball $B \in L_k$ contains $2^k$ leaf nodes. The choice of $k$ presents a trade-off: larger balls capture longer-range dependencies, while smaller balls are more resource-efficient. For each ball $B \in L_k$, we collect the leaf nodes within B:

$$\text{leaves}_B = \{B' \in L_{\text{leaf}} \mid B' \subset B\} \quad (6)$$

along with their features from $X_{\text{leaf}}$:

$$X_B = \{\mathbf{x}_{B'} \in X_{\text{leaf}} \mid B' \in \text{leaves}_B\} \quad (7)$$

We then compute self-attention independently on each ball[2]:

$$X'_B = \text{BAtt}(X_B) := \text{Att}(X_B \mathbf{W}_q, X_B \mathbf{W}_k, X_B \mathbf{W}_v) \quad (8)$$

where weights are shared between balls and the output $X'_B$ maintains row correspondence with $X_B$.

**Computational cost** As attention is computed independently for each ball $B \in L_k$, the computational cost is reduced from quadratic to linear. Precisely, for ball attention, the complexity is $\mathcal{O}(|B|^2 \cdot \frac{n}{|B|})$, i.e., quadratic in the ball size and linear in the number of balls:

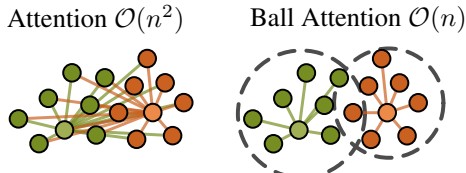

*Figure 4.* For highlighted points, standard attention computes interactions with all other points in the point cloud, while ball attention only considers points within their balls.

**Positional encoding** We introduce positional information to the attention layer in two ways. First, we augment the features of leaf nodes with their relative positions with respect to the ball's center of mass (relative position embedding):

$$\text{RPE}: \quad X_B = X_B + (P_B - \mathbf{c}_B)\mathbf{W}_{\text{pos}} \quad (9)$$

where $P_B$ contains positions of leaf nodes, $\mathbf{c}_B$ is the center of mass, and $\mathbf{W}_{\text{pos}}$ is a learnable projection. This allows the layer to incorporate geometric structure within each ball.

Second, we introduce a distance-based attention bias:

$$\mathcal{B}_B = -\sigma^2 ||\mathbf{c}_{B'} - \mathbf{c}_{B''}||_2, \quad B', B'' \in \text{leaves}_B \quad (10)$$

with a learnable parameter $\sigma \in \mathbb{R}$ (Wessels et al., 2024). The term decays rapidly as the distance between two nodes increases, which enforces locality and helps to mitigate potential artifacts from the tree building, particularly in cases where distant points are grouped together.

**Cross-ball connection** To increase the receptive field of our attention operator, we implement cross-ball connections inspired by the shifted window approach in Swin Transformer (Liu et al., 2021). There, patches are displaced diagonally by half their size to obtain two different image partitioning configurations. This operation can be equivalently interpreted as keeping the patches fixed while sliding the image itself.

---

[2]For any set of vectors $X$, we abuse notation by treating $X$ as a matrix with vectors as its rows.

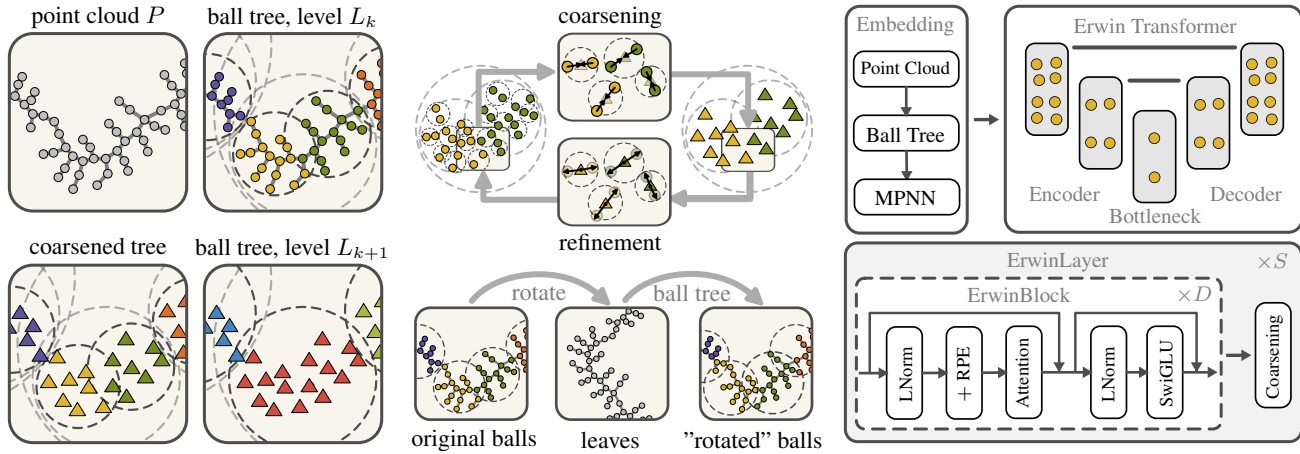

*Figure 3.* Overview of Erwin. **Left:** A sequence of two ball attention layers with intermediate tree coarsening. In every layer, attention is computed on partitions of size 16, which correspond to progressively higher levels of hierarchy. **Center (top):** Coarsening and refinement of a ball tree. **Center (bottom):** Building a tree on top of a rotated configuration for cross-ball interaction. **Right:** Architecture of Erwin.

Following this interpretation, we rotate the point cloud and construct the second ball tree $T_{\mathrm{rot}} = \{L_0^{\mathrm{rot}}, ..., L_m^{\mathrm{rot}}\}$, which induces a permutation $\pi^{\mathrm{rot}}$ of leaf nodes (see Fig. 3, center). We can then compute ball attention on the rotated configuration by first permuting the features according to $\pi^{\mathrm{rot}}$, applying attention, and then permuting back:

$$X_B' = \pi_{\mathrm{rot}}^{-1}\left(\mathrm{BAtt}\left(\pi_{\mathrm{rot}}\left(X_B\right)\right)\right) \tag{11}$$

By alternating between the original and rotated configurations in consecutive layers, we ensure the interaction between leaf nodes in otherwise separated balls.

**Tree coarsening/refinement** For larger systems, we are interested in coarser representations to capture features at larger scales. The coarsening operation allows us to hierarchically aggregate information by pooling leaf nodes to the centers of containing balls at $l$ levels higher (see Fig. 3, top, $l = 1$). Suppose the leaf level is $k$. For every ball $B \in L_{k+l}$, we concatenate features of all interior leaf nodes along with their relative positions with respect to $\mathbf{c}_B$ and project them to a higher-dimensional representation:

$$\mathbf{x}_B = \left(\bigoplus_{B' \in \mathrm{leaves}_B} [\mathbf{x}_{B'}, \mathbf{c}_{B'} - \mathbf{c}_B]\right)\mathbf{W}_c \tag{12}$$

where $\bigoplus$ denotes leaf-wise concatenation, and $\mathbf{W}_c \in \mathbb{R}^{C' \times 2^l(C+d)}$ is a learnable projection that increases the feature dimension to maintain expressivity. After coarsening, balls at level $k+l$ become the new leaf nodes, $L_{\mathrm{leaf}} := L_{k+l}$, with features $X_{\mathrm{leaf}} := \{\mathbf{x}_B \mid B \in L_{k+l}\}$. To highlight the simplicity of our method, we provide the pseudocode[3]:

---

[3]We use `einops` (Rogozhnikov, 2022) primitives.

```
# coarsening ball tree
x = rearrange([x, rel.pos], "(n 2^l) d → n (2^l d)") @ W_c
pos = reduce(pos, "(n 2^l) d → n d", "mean")
```

The inverse operation, refinement, allocates information from a coarse representation back to finer scales. More precisely, for a ball $B \in L_k$, its features are distributed back to the nodes at level $L_{k-l}$ contained within $B$ as:

$$\{\mathbf{x}_{B'} \mid B' \in L_{k-l}\} = [\mathbf{x}_B, P_B - \mathbf{c}_B]\,\mathbf{W}_r \tag{13}$$

where $P_B$ contains positions of all nodes at level $k-l$ within ball $B$ with center of mass $\mathbf{c}_B$, and $\mathbf{W}_r \in \mathbb{R}^{2^l C \times (C'+d)}$ is a learnable projection. After refinement, $L_{\mathrm{leaf}}$ and $X_{\mathrm{leaf}}$ are updated accordingly. In pseudocode:

```
# refining ball tree
x = [rearrange(x, "n (2^l d) → (n 2^l) d"), rel.pos] @ W_r
```

### 4.2. Model architecture

We are now ready to describe the details of the main model to which we refer as *Erwin*[4] (see Fig. 3) - a hierarchical transformer operating on ball trees.

**Embedding** At the embedding phase, we first construct a ball tree on top of the input point cloud and pad the leaf layer to complete the tree, as described in Section 3.2. To capture local geometric features, we employ a small-scale MPNN, which is conceptually similar to PointTransformer's embedding module using sparse convolution. When input connectivity is not provided (e.g., mesh), we utilize the ball tree structure for a fast nearest neighbor search.

---

[4]We pay homage to Swin Transformer as our model is based on rotating windows instead of sliding, hence Rwin → Erwin.

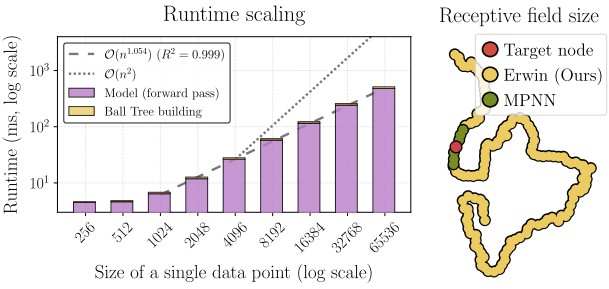

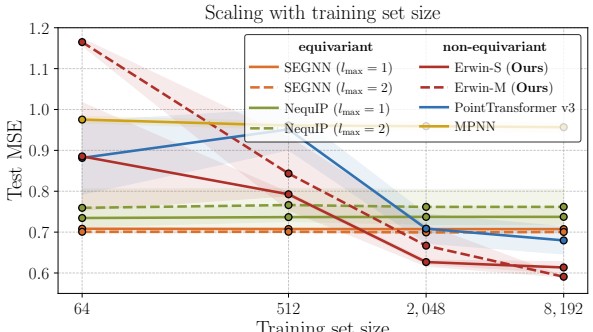

*Figure 5.* **Left:** Computational cost of Erwin. We split the total runtime into building a ball tree and running a model. The input is a batch of 16 point clouds, each of size $n$. We fit a power law which indicates close to linear scaling. **Right:** Receptive field of MPNN vs Erwin, $n = 800$. A node is in the receptive field if changing its features affects the target node's output. MPNN consists of 6 layers, each node connected to 16 nearest neighbours.

**ErwinBlock**    The core building block of Erwin follows a standard pre-norm transformer structure: LayerNorm followed by ball attention with a residual connection, and a SwiGLU feed-forward network (Shazeer, 2020). For the ball attention, the size $2^k$ of partitions is a hyperparameter. To ensure cross-ball interaction, we alternate between the original and rotated ball tree configurations, using an even number of blocks per **ErwinLayer** in our experiments.

**Overall architecture**    Following a UNet structure (Ronneberger et al., 2015; Wu et al., 2024b), Erwin processes features at multiple scales through encoder and decoder paths (Fig. 3, right). The encoder progressively coarsens the ball tree while increasing feature dimensionality to maintain expressivity. The coarsening factor is a hyperparameter that takes values that are powers of 2. At the decoder stage, the representation is refined back to the original resolution, with skip connections from corresponding encoder levels enabling multi-scale feature integration.

## 5. Experiments

The code is available at https://github.com/maxxxzdn/erwin. We summarize datasets in Table 1 and provide implementation details in Appendix B.

*Table 1.* Summary of benchmark datasets. † indicates varying size, for which we report the average number of nodes.

| GEOMETRY | BENCHMARKS | #DIM | #NODES |
|---|---|---|---|
| POINT CLOUD | ELASTICITY | 2D | 972 |
| | COSMOLOGY | 3D | 5,000 |
| | MD† | 3D+TIME | 890 |
| STRUCTURED | PLASTICITY | 2D+TIME | 3,131 |
| | AIRFOIL | 2D | 11,271 |
| | PIPE | 2D | 16,641 |
| UNSTRUCTURED | SHAPE-NET CAR | 3D | 32,186 |
| | EAGLE† | 2D+TIME | 3,388 |

*Figure 6.* Test mean-squared error (MSE) on the predicted velocities as a function of training set size for the cosmology task, 5 runs per point. Point transformers indicate favourable scaling surpassing graph-based models with sufficiently many training samples.

**Computational cost**    To experimentally evaluate Erwin's scaling, we learn the power-law[5] form $\mathrm{Runtime} = C \cdot n^\beta$ by first applying the logarithm transform to both sides and then using the least squares method to evaluate $\beta$. The result is an approximately linear scaling with $\beta{=}1.054$ with $R^2{=}0.999$, see Fig. 5. Ball tree construction accounts for $<5\%$ of the overall time (see Tables 7,8), proving the efficiency of our method for linearizing attention for point clouds.

**Receptive field**    One of the theoretical properties of our model is that with sufficiently many layers, its receptive field is global. To verify this claim experimentally, for an arbitrary target node, we run the forward pass of Erwin and MPNN and compute gradients of the node output with respect to all input nodes' features. If the gradient is non-zero, the node is considered to be in the receptive field of the target node. The visualization is provided in Fig. 5, right, where we compare the receptive field of our model with that of MPNN. As expected, the MPNN has a limited receptive field, as it cannot exceed $N$ hops, where $N$ is the number of message-passing layers. Conversely, Erwin implicitly computes all-to-all interactions, enabling it to capture long-range interactions in the data.

### 5.1. Cosmological simulations

To demonstrate our model's ability to capture long-range interactions, we use the cosmology benchmark (Balla et al., 2024), which consists of large-scale point clouds representing potential galaxy distributions.

**Dataset**    The dataset is derived from N-body simulations that evolve dark matter particles from the early universe to the present time. After the simulation, gravitationally bound structures (halos) are identified, from which the 5,000 heaviest ones are selected as potential galaxy locations. The halos form local clusters through gravity while maintaining

---

[5]We only use data for $n \geq 1024$ to exclude overhead costs.

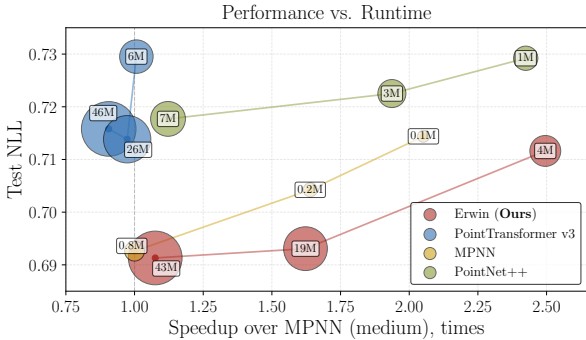

*Figure 7.* Test negative log-likelihood (NLL) of the predicted acceleration distribution for the molecular dynamics task (averaged over 3 runs). The baseline MPNN is taken from (Fu et al., 2022). The size of the markers reflects the number of parameters.

*Table 2.* Ablation on the cosmology task. Increasing window size improves performance at the cost of slower runtime (Erwin-S).

| BALL SIZE | 256 | 128 | 64 | 32 |
|---|---|---|---|---|
| TEST LOSS | **0.595** | 0.603 | 0.612 | 0.620 |
| RUNTIME, MS | 229.6 | 165.2 | 135.3 | **126.0** |

*Table 3.* Ablation study on architectural choices: using MPNN in embedding, RPE, and cross-ball connection via rotating trees.

| MODEL | TEST LOSS | |
|---|---|---|
| | MD | SHAPENET-CAR |
| W/O | 0.738 | 30.39 |
| + MPNN | 0.720 | 30.49 |
| + RPE | 0.715 | 30.02 |
| + ROTATING TREE | 0.712 | 15.85 |

long-range correlations that originated from interactions in the early universe, reflecting the initial conditions.

**Task** The input is a point cloud $\mathbf{X} \in \mathbb{R}^{5000 \times 3}$, where each row corresponds to a galaxy and each column to $x, y, z$ coordinate respectively. The task is a regression problem to predict the velocity of every galaxy $\mathbf{Y} \in \mathbb{R}^{5000 \times 3}$. We vary the size of the training dataset from 64 to 8,192, while the validation and test datasets have a fixed size of 512. The models are trained using the mean squared error between predicted and ground truth velocities.

**Results** The results are shown in Fig. 6. We compare against multiple equivariant (NequIP (Batzner et al., 2021), SEGNN (Brandstetter et al., 2022)) and non-equivariant (MPNN (Gilmer et al., 2017), PTv3 (Wu et al., 2024b)) baselines. In the small data regime, graph-based equivariant models are preferable. However, as the training set size increases, their performance plateaus. We note that this is also the case for non-equivariant MPNNs, suggesting the issue might arise from failing to capture medium to large-scale interactions, where increased local expressivity has minimal impact. Conversely, transformer-based models scale favorably with training set size and eventually surpass graph-based models, highlighting their ability to capture both small and large-scale interactions. Our model demonstrates particularly strong performance and significantly outperforms other baselines for larger training sets.

### 5.2. Molecular dynamics

Molecular dynamics (MD) is essential for understanding physical and biological systems at the atomic level but remains computationally expensive even with neural network potentials due to all-atom force calculations and femtosecond timesteps required to maintain stability and accuracy. Fu et al. (2022) suggested accelerating MD simulation through coarse-grained dynamics with an MPNN. In this experiment, we take a different approach and instead oper-

ate on the original representation but improve the runtime by employing our hardware-efficient model. Therefore, the question we ask is how much we can accelerate a simulation w.r.t. an MPNN without compromising the performance.

**Dataset** The dataset consists of single-chain coarse-grained polymers (Webb et al., 2020; Fu et al., 2022) simulated using MD. Each system includes 4 types of coarse-grained beads interacting through bond, angle, dihedral, and non-bonded potentials. The training set consists of polymers with repeated bead patterns while the test set polymers are constructed by randomly sampling bead sequences, thus introducing a challenging distribution shift. The training set contains 100 short trajectories (50k $\tau$), while the test set contains 40 trajectories that are 100 times longer.

**Task** We follow the experimental setup from Fu et al. (2022). The model takes as input a polymer chain of $N$ coarse-grained beads. Each bead has a specific weight and is associated with the history $\{\dot{\mathbf{x}}_{t-16\Delta t}, ..., \dot{\mathbf{x}}_{t-\Delta t}\}$ of (normalized) velocities from 16 previous timesteps at intervals of $\Delta t = 5\tau$. The model predicts the mean $\mu_t \in \mathbb{R}^{N \times 3}$ and variance $\sigma_t^2 \in \mathbb{R}_+^{N \times 3}$ of (normalized) acceleration for each bead, assuming a normal distribution. We train using the negative log-likelihood loss between predicted and ground truth accelerations computed from the ground truth trajectories.

**Results** The results are given in Fig. 7. As baselines, we use MPNN (Gilmer et al., 2017) as well as two hardware-efficient architectures: PointNet++ (Qi et al., 2017) and PTv3 (Wu et al., 2024b). Notably, model choice has minimal impact on performance, potentially due to the absence of long-range interactions as the CG beads do not carry any charge. Furthermore, it is sufficient to only learn local bonded interactions. There is, however, a considerable improvement in runtime for Erwin (1.7-2.5 times depending on the size), which is only matched by a smaller MPNN or PointNet++, both having significantly higher test loss.

*Table 4.* RMSE on PDE benchmarks from Li et al. (2023b). Transformer-based models are taken as baselines from Luo et al. (2025).

| MODEL | RMSE | | | |
|---|---|---|---|---|
| | ELASTICITY | PLASTICITY | AIRFOIL | PIPE |
| LNO (2024) | 0.69 | 0.29 | 0.53 | 0.31 |
| GALERKIN (2021) | 2.40 | 1.20 | 1.18 | 0.98 |
| HT-NET (2022) | / | 3.33 | 0.65 | 0.59 |
| OFORMER (2023C) | 1.83 | 0.17 | 1.83 | 1.68 |
| GNOT (2023) | 0.86 | 3.36 | 0.76 | 0.47 |
| FACTFORMER (2023D) | / | 3.12 | 0.71 | 0.60 |
| ONO (2024) | 1.18 | 0.48 | 0.61 | 0.52 |
| TRANSOLVER++ (2025) | 0.52 | 0.11 | **0.48** | **0.27** |
| **ERWIN (OURS)** | **0.34** | **0.10** | 2.57 | 0.61 |

*Table 5.* Test MSE for ShapeNet-Car pressure prediction. Baseline results are taken from Bleeker et al. (2025).

| MODEL | MSE |
|---|---|
| POINTNET (2017) | 43.36 |
| GINO (2023B) | 35.24 |
| UPT (2024A) | 31.66 |
| TRANSOLVER (2024A) | 19.88 |
| GP-UPT (2025) | 17.02 |
| PTv3-S (2024B) | $19.09 \pm 0.67$ |
| PTv3-M (2024B) | $17.42 \pm 0.38$ |
| **ERWIN-S (OURS)** | $15.85 \pm 0.19$ |
| **ERWIN-M (OURS)** | $15.43 \pm 0.45$ |

### 5.3. PDE benchmarks and airflow pressure

Deep learning models have emerged as surrogate solvers of partial differential equations (PDEs), learning to approximate solutions from data (Li et al., 2021; Wu et al., 2024a). While their advantages over traditional numerical methods remain unclear (McGreivy & Hakim, 2024), the task itself serves as a surrogate for large-scale applications like weather forecasting (Price et al., 2025) and fluid dynamics (Bleeker et al., 2025), where conventional solvers become computationally prohibitive. Furthermore, in this task, we are interested in the model's ability to scale to large domains while capturing complex patterns in underlying physics.

**Dataset** We benchmark on multiple datasets taken from Li et al. (2023a). Each dataset is defined either on point cloud (Elasticity) or structured mesh (Plasticity, Airfoil, Pipe). Additionally, we evaluate our model on airflow pressure modeling (Umetani & Bickel, 2018; Alkin et al., 2024a). It consists of 889 car models, each car represented by 3,586 surface points in 3D space. Airflow was simulated around each car for 10s ($Re = 5 \times 10^6$) and averaged over the last 4 s to obtain pressure values at each point.

**Task** For the PDE benchmarks, we follow the pipeline from Wu et al. (2024a) and minimize the relative L2 error; see Appendix B for details. For ShapeNet-Car, the task is to estimate the surface pressure $\mathbf{Y} \in \mathbb{R}^{3388 \times 1}$ given surface points $\mathbf{X} \in \mathbb{R}^{3388 \times 3}$. We train by optimizing the mean squared error between predicted and ground truth pressure.

**PDE benchmarks** The results are given in Table 4, where we compare against other transformer-based methods. Erwin achieves state-of-the-art performance on 2 out of 4 tasks. Interestingly, it dramatically underperforms on the Airfoil task, which indicates a failure mode. We speculate that this is related to a specific structure of the data - the density of the mesh decreases dramatically moving away from the center of mass. This means that points across different balls have considerably varying density, which poses a challenge.

**ShapeNet-Car** See Table 5 for results. Both Erwin and PTv3 achieve significantly lower test MSE compared to other models[6]. We note that the best performing configuration of Erwin and PTv3 did not include any coarsening, thus operating directly on the original point cloud. This indicates that the task favors the ability of a model to capture fine geometric details. In comparison, other approaches introduce information loss through compression - UPT (Alkin et al., 2024a) and Transolver (Wu et al., 2024a) involve pooling to the latent space, while GINO (Li et al., 2023b) interpolates the geometry onto regular grids and back.

### 5.4. Turbulent fluid dynamics

In the last experiment, we demonstrate the expressivity of our model by simulating turbulent fluid dynamics. The problem is notoriously challenging due to multiple factors: the inherently nonlinear behavior of fluids, the multiscale and chaotic nature of turbulence, and the presence of long-range dependencies. Moreover, the geometry of the simulation domain and the presence of objects introduce complex boundary conditions, thus adding another layer of complexity.

**Dataset** We use EAGLE (Janny et al., 2023), a large-scale benchmark of unsteady fluid dynamics. Each simulation includes a flow source (drone) that moves in 2D environments with different boundary geometries, producing airflow. The time evolution of velocity and pressure fields is recorded along with dynamically adapting meshes. The dataset contains 600 different geometries of 3 types, with approximately 1.1 million 2D meshes averaging 3,388 nodes each. The total dataset includes 1,184 simulations with 990 time steps per simulation. The dataset is split with 80% for training and 10% each for validation and testing.

**Task** We follow the original experimental setup of the benchmark. The input is the velocity $V \in \mathbb{R}^{N \times 2}$ and pres-

---

[6]The baseline results for the ShapeNet-Car task, except for PTv3, were taken from Bleeker et al. (2025).

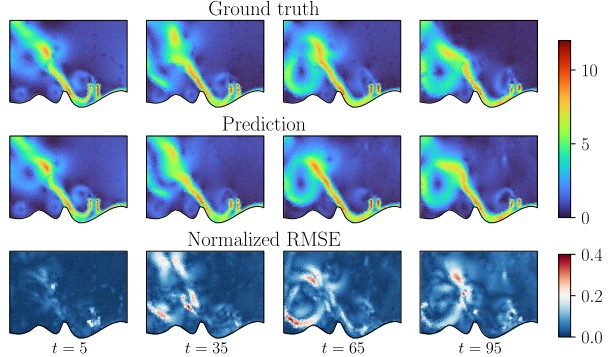

Ground truth

Prediction

Normalized RMSE

$t = 5$     $t = 35$     $t = 65$     $t = 95$

*Figure 8.* The norm of the velocity field at different steps of the rollout trajectories.

sure $P \in \mathbb{R}^{N \times 2}$ fields evaluated at every node of the mesh at time step $t$, along with the type of the node. The task is to predict the state of the system at the next time step $t + 1$. The training is done by predicting a trajectory of states of length 5 and optimizing the loss

$$\mathcal{L} = \sum_{i=1}^{5} \Big( \mathrm{MSE}(V_{t+i}, \hat{V}_{t+i}) + \alpha \, \mathrm{MSE}(P_{t+i}, \hat{P}_{t+i}) \Big),$$

where $\alpha = 0.1$ is the parameter that balances the importance of the pressure field over the velocity field.

**Results** For comparison, we include the baselines from the original benchmark: MeshGraphNet (MGN; Pfaff et al., 2021), GAT (Velickovic et al., 2018), DilResNet (DRN; Stachenfeld et al., 2021) and EAGLE (Janny et al., 2023)[7]. The first two baselines are based on message-passing, while DilResNet operates on regular grids, hence employing interpolation for non-uniform meshes. EAGLE uses message-passing to pool the mesh to a coarser representation with a fixed number of clusters, on which attention is then computed. The quantitative results are given in Table 6 and unrolling trajectories are shown in Fig. 8, 9. Erwin demonstrates strong results on the benchmark and outperforms every baseline, performing especially well at predicting pressure. In terms of inference time and memory consumption, Erwin achieves substantial gains over EAGLE, being 3 times faster and using 8 times less memory.

### 5.5. Ablation study

We also conducted an ablation study to examine the effect of increasing ball sizes on the model's performance in the cosmology experiment; see Table 2. Given the presence of long-range interactions in the data, larger window sizes (and thus receptive fields) improve model performance, albeit at the cost of increased computational runtime. Our architectural ablation study on the MD task (Table 3) reveals that using an MPNN at the embedding step produces substantial

[7]We additionally trained UPT (Alkin et al., 2024a), but were not able to obtain competitive results in our initial experiments.

*Table 6.* RMSE on velocity V and pressure P fields across different prediction horizons (mean ± std over 5 runs). Inference runtime and memory use are computed for a batch of 8, avg. 3,500 nodes.

| HORIZON | +1 | | +50 | | TIME | MEM. |
|---|---|---|---|---|---|---|
| FIELD / UNIT | V | P | V | P | (MS) | (GB) |
| MGN | 0.081 | 0.43 | 0.592 | 2.25 | 40 | 0.7 |
| GAT | 0.170 | 64.6 | 0.855 | 163 | 44 | 0.5 |
| DRN | 0.251 | 1.45 | 0.537 | 2.46 | 42 | **0.2** |
| EAGLE | 0.053 | 0.46 | 0.349 | 1.44 | 30 | 1.5 |
| **ERWIN** | **0.044** | **0.31** | **0.281** | **1.15** | **11** | **0.2** |
| (**OURS**) | ±0.001 | ±0.01 | ±0.001 | ±0.06 | | |

improvements, likely due to its effectiveness in learning local interactions. At the same time, that did not generalize to ShapeNet-Car, where the most decisive factor was including cross-ball interactions, highlighting the importance of capturing fine-grained details in this task.

## 6. Conclusion

We present Erwin, a hierarchical transformer that uses ball tree partitioning to process large-scale physical systems with linear complexity. Erwin achieves state-of-the-art performance in cosmology, turbulent fluid dynamics, and, partially, on standard PDE benchmarks, demonstrating its effectiveness across diverse physical domains. The efficiency of Erwin makes it a suitable candidate for tasks that require modeling large particle systems, such as computational chemistry (Fu et al., 2024) or diffusion-based molecular dynamics (Jing et al., 2024).

**Limitations and Future Work** Because Erwin relies on perfect binary trees, we need to pad the input set with virtual nodes, which induces computational overhead for ball attention computed over non-coarsened trees (first ErwinBlock). This issue can be circumvented by employing learnable pooling to the next level of the ball tree, which is always full, ensuring the remaining tree is perfect. Whether we can perform such pooling without sacrificing expressivity is a question that we leave to future research.

Erwin relies on cross-ball interaction and coarsening to capture long-range interactions. Both mechanisms have inherent limitations: the former requires multiple steps for signals to propagate between balls, while the latter sacrifices fine-grained detail. A promising approach to address these issues is adapting sparse attention methods such as Native Sparse Attention (Yuan et al., 2025). This framework aligns naturally with Erwin's ball tree structure and would enable learning distant interactions while preserving full resolution. Finally, Erwin is neither permutation nor rotation equivariant, although rotation equivariance can be incorporated without compromising scalability, such as via Geometric Algebra Transformers (Brehmer et al., 2023) or Fast Euclidean Attention (Frank et al., 2024).

## Acknowledgements

We are grateful to Evgenii Egorov and Ana Lučić for their feedback and inspiration. MZ acknowledges support from Microsoft Research AI4Science. JWvdM acknowledges support from the European Union Horizon Framework Programme (Grant agreement ID: 101120237).

## Impact statement

The broader implications of our work are primarily in moderate- to large-scale scientific applications, such as molecular dynamics or computational fluid dynamics. We believe that efficient and expressive architectures like Erwin could become a foundation for resource-intensive deep learning frameworks and therefore help in better understanding the physical systems governing our world.

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

**Algorithm 1** BUILDBALLTREE

    **input** Array of data points $D$ in $\mathbb{R}^d$
    **output** Ball tree node $B$

    **if** $|D| = 1$ **then**
        Create leaf node $B$ containing single point in $D$
        **return** $B$
    **end if**

    # Find dimension of greatest spread
    $\delta \leftarrow \text{argmax}_{i \in 1, \ldots, d}(\max_{x \in D} x_i - \min_{x \in D} x_i)$

    # Find the median point along $\delta$
    $p \leftarrow \text{median}\{x_\delta \mid x \in D\}$
    # Points left of median along $\delta$
    $L \leftarrow \{x \in D \mid x_\delta \leq p_\delta\}$
    # Points right of median along $\delta$
    $R \leftarrow \{x \in D \mid x_\delta > p_\delta\}$

    # Recursively construct children
    $B.\text{child}_1 \leftarrow \text{BUILDBALLTREE}(L)$
    $B.\text{child}_2 \leftarrow \text{BUILDBALLTREE}(R)$

    **return** $B$

## A. Implementation details

**Ball tree construction**    The algorithm used for constructing ball trees (Pedregosa et al., 2012) can be found in Alg. 1. Note that this implementation is not rotationally equivariant as it relies on choosing the dimension of the greatest spread, which in turn depends on the original orientation. Examples of ball trees built in our experiments are shown in Fig. 9.

**MPNN in the embedding**    Erwin employs a small-scale MPNN in the embedding. More precisely, given a graph $G = (V, E)$ with nodes $v_i \in V$ and edges $e_{ij} \in E$, we compute multiple layers of message-passing as proposed in (Gilmer et al., 2017):

$$\mathbf{m}_{ij} = \text{MLP}_e(\mathbf{h}_i, \mathbf{h}_j, \mathbf{p}_i - \mathbf{p}_j), \qquad \text{message}$$

$$\mathbf{m}_i = \sum_{j \in \mathcal{N}(i)} \mathbf{m}_{ij}, \qquad \text{aggregate} \quad (14)$$

$$\mathbf{h}_i = \text{MLP}_h(\mathbf{h}_i, \mathbf{m}_i), \qquad \text{update}$$

where $\mathbf{h}_i \in \mathbb{R}^H$ is a feature vector of $v_i$, and $\mathcal{N}(i)$ denotes the neighborhood of $v_i$. The motivation for using an MPNN is to incorporate local neighborhood information into the model. Theoretically, attention should be able to capture this information as well; however, this might require substantially increasing the feature dimension and the number of attention heads, which would be prohibitively expensive for a large number of nodes in the original level of a ball tree. In our experiments, we consistently maintain the size of $\text{MLP}_e$ and $\text{MLP}_h$ to be small ($H \leq 32$) such that embedding accounts for less than 5% of total runtime.

## B. Experimental details

In this section, we provide comprehensive experimental details regarding hardware specifications, hyperparameter choices, and optimization procedures.

### B.1. Hardware and Software

All experiments were conducted on a single NVIDIA RTX A6000 GPU with 48GB memory and 16 AMD EPYC™ 7543 CPUs. Erwin and all baselines except those for cosmology were implemented in PyTorch 2.6. For the cosmology benchmark, SEGNN, NequIP, and MPNN baselines were implemented in JAX as provided by the original benchmark repository. Training times for Erwin varied by task: cosmology (5-10 minutes depending on training set size), molecular dynamics (2-4 hours depending on model size), PDE benchmarks (8 hours for Elasticity, 48 hours for others), ShapeNet-Car (2 hours), and EAGLE (48 hours).

### B.2. Training Details

All models were trained using the AdamW optimizer (Loshchilov & Hutter, 2019) with weight decay $10^{-5}$. The learning rate was tuned in the range $10^{-4}$ to $10^{-3}$ to minimize loss on the respective validation sets with cosine decay to $10^{-7}$. Gradient clipping by norm with value 1.0 was applied across all experiments. Early stopping was used only for ShapeNet-Car and molecular dynamics tasks, while all other models were trained until convergence. In every experiment, we normalize inputs to the model. Hyperparameter optimization was performed using grid search with single trials.

### B.3. Dataset Splits and Evaluation

Dataset splits followed the original benchmarks:

- **Cosmology**: Training set varied from 64 to 8192 examples, with validation and test sets of 512 examples each

- **Molecular Dynamics**: 100 short trajectories for training, 40 long trajectories for testing

- **PDE Benchmarks**: 1000 training / 200 test examples (except Plasticity: 900/80)

- **ShapeNet-Car**: 700 training / 189 test examples

- **EAGLE**: 1184 trajectories with 80%/10%/10% split

For statistical significance, we ran each experiment 5 times and report mean and standard deviation for cosmology, molecular dynamics, ShapeNet-Car, and EAGLE. PDE experiments were run once due to computational constraints.

ball size 512           ball size 256           ball size 128

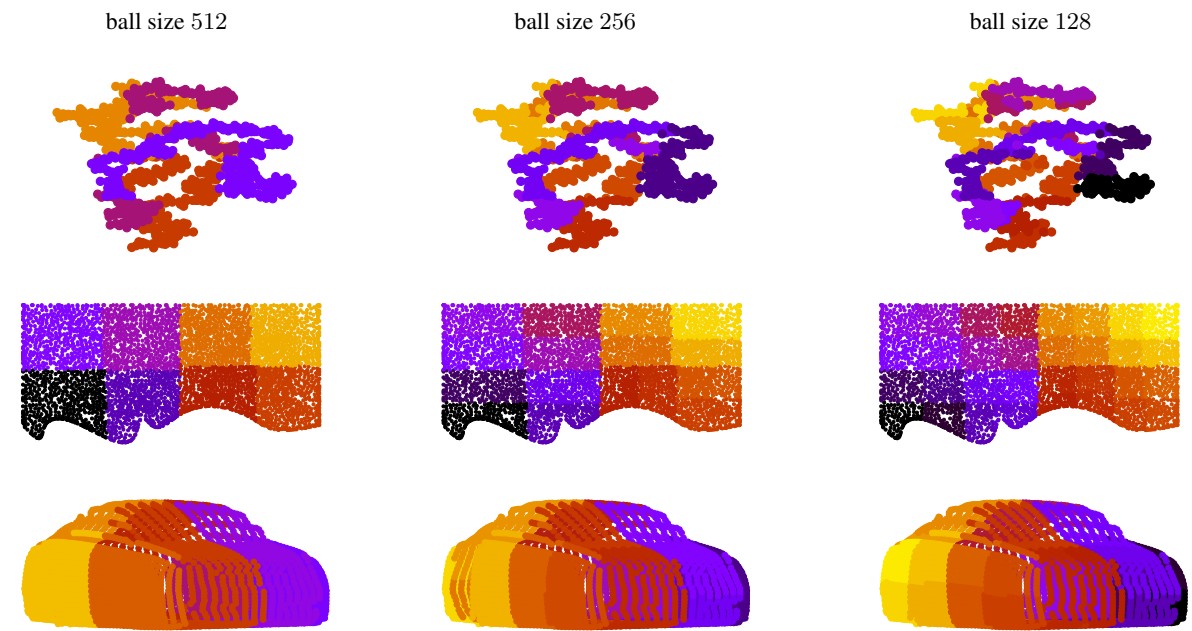

*Figure 9.* Examples of ball trees built on top of data. Partitions at different levels of ball trees are shown. **Top:** A polypeptide from the molecular dynamics task. **Center:** A domain from the EAGLE dataset. **Bottom:** A car surface from the ShapeNet-Car dataset.

### B.4. Evaluation Metrics

For RMSE computation, we use the relative L2 error: RMSE := $\|f(x) - y\|/\|y\|$. For molecular dynamics, we randomly sample 16 history points from each trajectory and predict one future point. Acceleration is predicted by the neural network based on history and compared against ground truth computed using forward differences. Model performance is evaluated using negative log-likelihood loss between predicted and ground truth accelerations.

### B.5. Baseline Implementations

All baseline results were taken from official implementations or reported values:

- **Cosmology**: Official JAX implementation (Balla et al., 2024), code from `https://github.com/smsharma/eqnn-jax`
- **ShapeNet-Car**: Results from Bleeker et al. (2025), code from `https://github.com/ml-jku/UPT`
- **PDE Benchmarks**: Results from Luo et al. (2025), `https://github.com/thuml/Transolver`
- **EAGLE**: Original implementation (Janny et al., 2023), `https://github.com/eagle-dataset/EagleMeshTransformer`
- **Molecular Dynamics**: MPNN and PointNet++ implemented by us, other models taken from official codebases.
- **PTv3**: `https://github.com/Pointcept/PointTransformerV3`

### B.6. Computational Efficiency

Table 7 shows the inference time breakdown for Erwin on NVIDIA RTX A6000 with batch size 16. Ball tree construction (Table 8) consistently accounts for less than 5% of total runtime, demonstrating the efficiency of our optimized implementation compared to standard libraries.

*Table 7.* Erwin runtime with `torch.compile`

| NODES PER BATCH | RUNTIME (MS) | | | |
|---|---|---|---|---|
| 16 × | 2048 | 4096 | 8192 | 16384 |
| FWD | 17.3 | 31.6 | 79.7 | 189 |
| FWD + BWD | 26.4 | 45.4 | 114 | 232 |

*Table 8.* Ball tree construction on 16 AMD EPYC™ 7543 CPUs.

| NODES PER BATCH, | RUNTIME (MS) | | | |
|---|---|---|---|---|
| 16 × | 2048 | 4096 | 8192 | 16384 |
| SKLEARN + JOBLIB | 16.3 | 21.2 | 24.1 | 44.0 |
| **OURS** | 0.73 | 1.54 | 3.26 | 6.98 |
| SPEED-UP | 22.3× | 13.8× | 7.4× | 6.3× |

### B.7. Further details per experiment

**Cosmological simulations** We follow the experimental setup of the benchmark. The training was done for 5,000 epochs with batch size 16 for point transformers and batch size 8 for message-passing-based models. The implementation of SEGNN, NequIP, and MPNN was done in JAX and

taken from the original benchmark repository (Balla et al., 2024). We maintained the hyperparameters of the baselines used in the benchmark. For Erwin and PTv3, the hyperparameters are provided in Table 9. In Erwin's embedding, we conditioned messages on Bessel basis functions rather than the relative position, which significantly improved overall performance.

**Molecular dynamics**    All models were trained with batch size 32 for 50,000 training iterations with an initial learning rate of $5 \cdot 10^{-4}$. We fine-tuned the hyperparameters of every model on the validation dataset (reported in Table 11).

**PDE benchmarks**    The baseline results and experimental setups are taken from Luo et al. (2025). We adjusted batch size, ball size, and the number of attention heads per block for the best performance. The hidden dimensionality of Erwin was adjusted such that the overall number of parameters is around $10^6$, which is comparable with other baselines. Constrained by the parameter size, the same configuration worked the best; see Table 12 for details.

**Airflow pressure modeling**    We take the results of baseline models from Bleeker et al. (2025). Both Erwin and PTv3 were trained with batch size 32 for 1,000 epochs, and their hyperparameters are given in Table 10. We tuned the number of blocks, the number of message-passing steps, hidden dimensionality, and ball size per block for the best performance. When experimenting with pooling, we found that not involving any coarsening significantly improves model performance; hence, we used stride 1 in each block.

**Turbulent fluid dynamics**    Baseline results are taken from (Janny et al., 2023), except for runtime and peak memory usage, which we measured ourselves. Erwin was trained with batch size 12 for 4,000 epochs. We tuned hidden dimensionality and ball size per block for the lowest validation loss.

*Table 9.* Model architectures for the cosmological simulations task. For varying sizes of Erwin, the values are given as (S/M).

| Model | Parameter | Value |
|-------|-----------|-------|
| PTv3 | Grid size | 0.01 |
| | Enc. depths | (2, 2, 6, 2) |
| | Enc. channels | (32, 64, 128, 256) |
| | Enc. heads | (2, 4, 8, 16) |
| | Enc. patch size | 64 |
| | Dec. depths | (2, 2, 2) |
| | Dec. channels | (64, 64, 128) |
| | Dec. heads | (2, 4, 8) |
| | Dec. patch size | 64 |
| | Pooling | (2, 2, 2) |
| Erwin | MPNN dim. | 32 |
| | Channels | 32-512/64-1024 |
| | Window size | 64 |
| | Enc. heads | (2, 4, 8, 16) |
| | Enc. depths | (2, 2, 6, 2) |
| | Dec. heads | (2, 4, 8) |
| | Dec. depths | (2, 2, 2) |
| | Pooling | (2, 2, 2, 1) |

*Table 11.* Model architectures for the molecular dynamics task. For models of varying sizes, the values are given as (S/M/L).

| Model | Parameter | Value |
|-------|-----------|-------|
| MPNN | Hidden dim. | 48/64/128 |
| | MP steps | 6 |
| | MLP layers | 2 |
| | Message agg-n | mean |
| PointNet++ | Hidden dim. | 64/128/196 |
| | MLP layers | 2 |
| PTv3 | Grid size | 0.025 |
| | Enc. depths | (2, 2, 2, 6, 2) |
| | Enc. channels | 16-192/24-384/64-1024 |
| | Enc. heads | (2, 4, 8, 16, 32) |
| | Enc. patch size | 128 |
| | Dec. depths | (2, 2, 2, 2) |
| | Dec. channels | 16-96/48-192/64-512 |
| | Dec. heads | (4, 4, 8, 16) |
| | Dec. patch size | 128 |
| Erwin | MPNN dim. | 16/16/32 |
| | Channels | (16-256/32-512/64-1024) |
| | Window size | 128 |
| | Enc. heads | (2, 4, 8, 16, 32) |
| | Enc. depths | (2, 2, 2, 6, 2) |
| | Dec. heads | (4, 4, 8, 16) |
| | Dec. depths | (2, 2, 2, 2) |
| | Pooling | (2, 2, 2, 2, 1) |

*Table 10.* Model architectures for the airflow pressure task.

| Model | Parameter | Value |
|-------|-----------|-------|
| PTv3 | Grid size | 0.01 |
| | Enc. depths | (2, 2, 2, 2, 2) |
| | Enc. channels | 24-384 |
| | Enc. heads | (2, 4, 8, 16, 32) |
| | Enc. patch size | 256 |
| | Dec. depths | (2, 2, 2, 2) |
| | Dec. channels | 48-192 |
| | Dec. heads | (4, 4, 8, 16) |
| | Dec. patch size | 256 |
| Erwin | MPNN dim. | 8 |
| | Channels | 96 |
| | Window size | 256 |
| | Enc. heads | (8, 16) |
| | Enc. depths | (6, 2) |
| | Dec. heads | (8,) |
| | Dec. depths | (2,) |
| | Pooling | (2, 1) |
| | MP steps | 1 |

*Table 12.* Model architectures for the PDE benchmarks.

| Model | Parameter | Value |
|-------|-----------|-------|
| Erwin | MPNN dim. | 64 |
| | Channels | 64 |
| | Window size | 256 |
| | Enc. heads | (8, 8) |
| | Enc. depths | (6, 6) |
| | Dec. heads | (8, 8) |
| | Dec. depths | (6) |
| | Pooling | (1, 1) |

*Figure 10.* The norm of the velocity field at different steps of the rollout trajectories, predicted by Erwin.

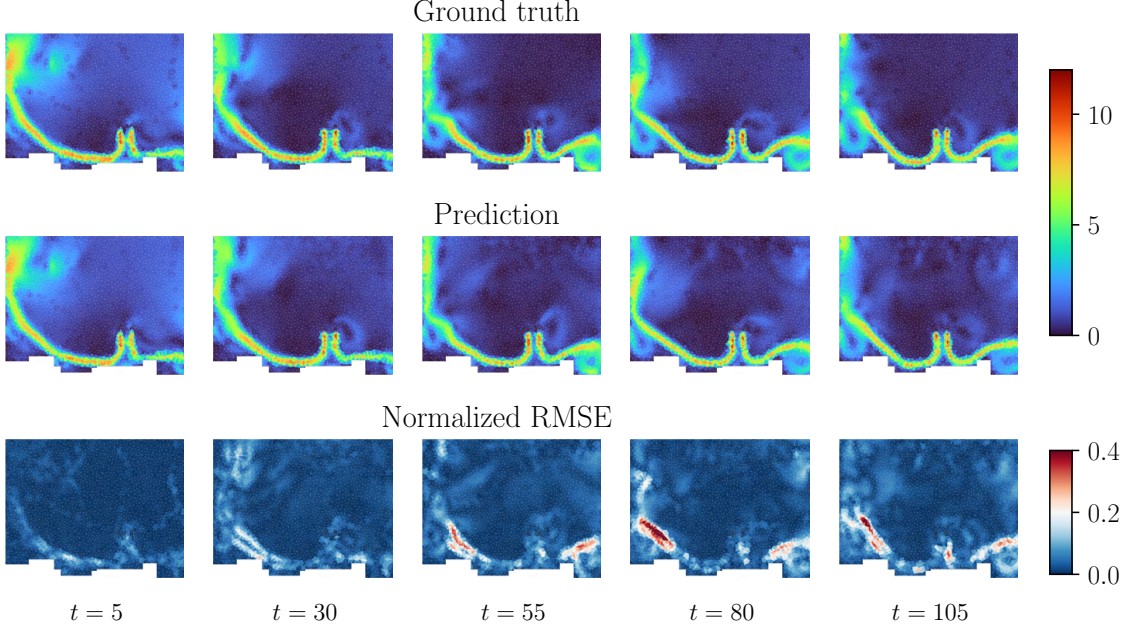

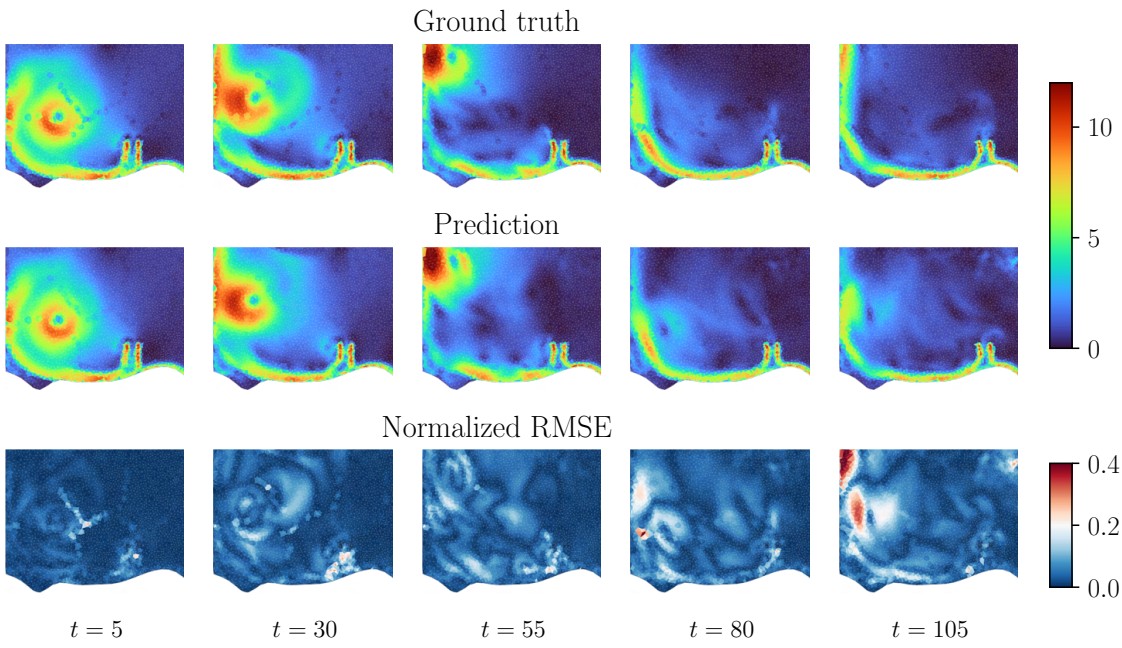

*Figure 11.* The norm of the pressure field at different steps of the rollout trajectories, predicted by Erwin.

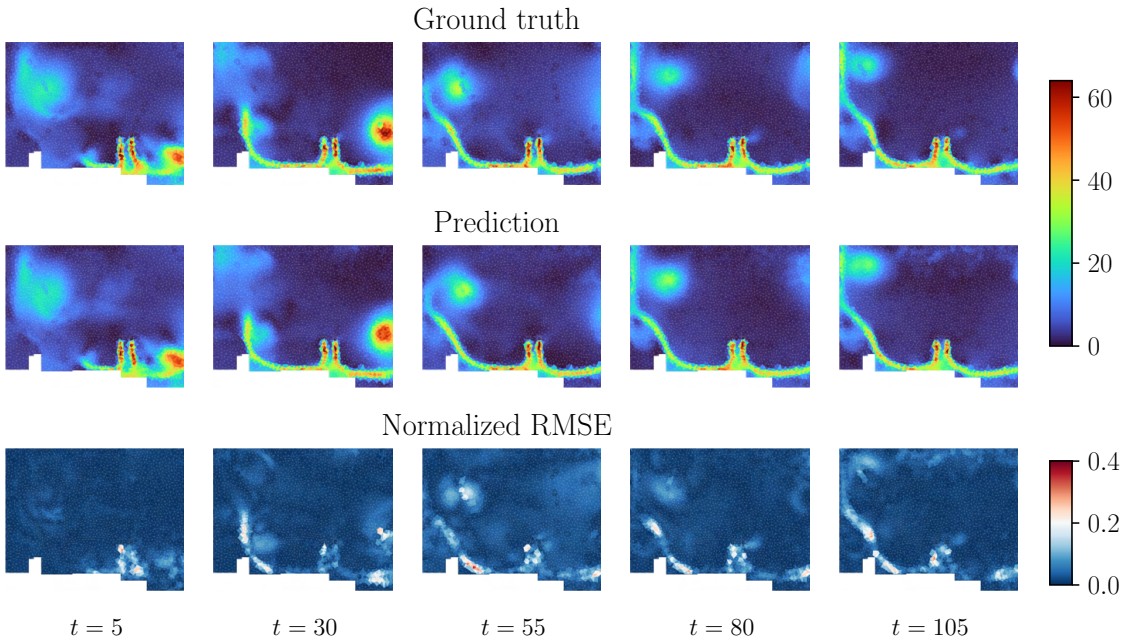

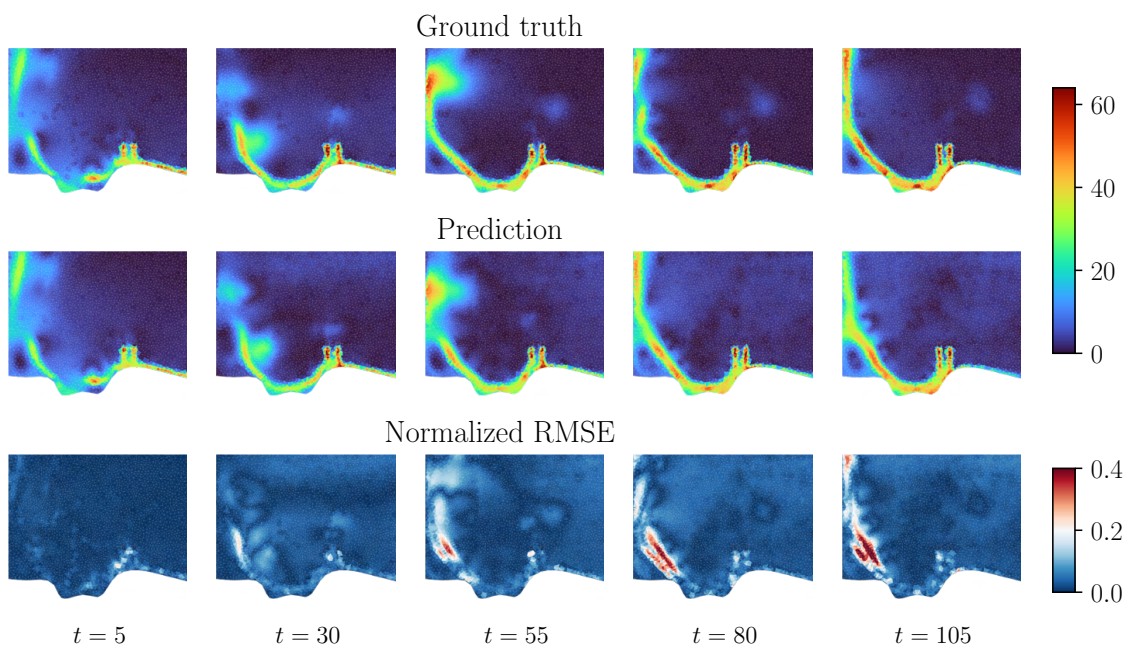

