# OpenReview forum: "Erwin: A Tree-based Hierarchical Transformer for Large-scale Physical Systems"
_ICML.cc/2025/Conference — ICML 2025 poster_

### Official Review · Reviewer_CTuS · 2025-03-04

**Overall Recommendation:** 3

**Summary:**

**Review updates after rebuttal**
The authors correspond to many of my commets well. As my initial score is already the positive side and I find it fair, hence I will keep my score unchanged.
**End Review updates after rebuttal**

The paper introduces Erwin, a hierarchical transformer that combines ball tree partitioning with attention mechanisms to efficiently process large-scale physical systems defined on irregular grids. Erwin achieves linear-time attention by organizing computation hierarchically, enabling it to capture both fine-grained local details and global features. The method is validated across multiple domains (cosmology, molecular dynamics, and turbulent fluid dynamics), demonstrating state-of-the-art performance in both accuracy and computational efficiency.

However, the paper has several areas that need improvement (details see in later sections):

- No comprehensive reports for baselines;

- Incomplete dataset descriptions;

- Lack of literature review for multiscale GNNs for Irregular Grids;

- Rushed presentation;

Overall, I give weak accept at current phase; upon resolving the above bullet points, I will improve my score.

**Claims And Evidence:**

Claim: Erwin achieves linear-time attention and outperforms baseline methods in accuracy and computational efficiency.

Evidence:

Figure 5~7: Shows linear scaling of runtime, lower MSE, and faster runtime compared to baselines
Table 3, 4: lower RMSE and MSE on two datasets.

Claim: Erwin captures long-range interactions and multi-scale phenomena effectively.

Evidence:

Figure 6~8

**Essential References Not Discussed:**

Multiscale learning on irregular mesh
- Multi-scale rotation-equivariant graph neural networks for unsteady Eulerian fluid dynamics
- Efficient Learning of Mesh-Based Physical Simulation with Bi-Stride Multi-Scale Graph Neural Network
- Learning Distributions of Complex Fluid Simulations with Diffusion Graph Networks

**Experimental Designs Or Analyses:**

The paper includes experiments across multiple domains (cosmology, molecular dynamics, fluid dynamics, airflow pressure), demonstrating the generalizability of Erwin.

The results show consistent improvements in both accuracy and computational efficiency.

Weaknesses:

Not all baselines are applied to all datasets. For example, PointTransformer v3 is not included in the fluid dynamics experiments, and EAGLE is not included in the cosmology experiments.

Suggestion: Include a comprehensive table listing all models and baselines for each dataset, with explanations for missing entries (e.g., due to incompatibility or heavy modifications required to the source).

**Methods And Evaluation Criteria:**

Methods:

- Ball tree partitioning to organize hierarchy.

- UNet-like structure, or Pointnet++, SWIN transformer, etc.

Evaluation Criteria:

Accuracy: Measured by MSE (cosmology, airflow pressure), NLL (molecular dynamics), and RMSE (fluid dynamics).

Efficiency: Measured by runtime and memory usage.

Scalability: Demonstrated through linear scaling of runtime with the number of nodes.

**Other Comments Or Suggestions:**

NA

**Other Strengths And Weaknesses:**

Weaknesses:

- Incomplete sections: Dataset introduction, experimental designs, baseline comparisons, and supplementary material need refinement.

- Rushed presentation: The paper feels slightly rushed, with some sections (e.g., dataset descriptions) lacking detail. Many tables in appendix lack text accompanying them.

- Suggestion: Revise the paper to address these issues, ensuring completeness and coherence.

**Questions For Authors:**

NA

**Relation To Broader Scientific Literature:**

The paper builds on hierarchical attention methods and tree-based algorithms from computational physics. It also draws inspiration from vision transformers (e.g., SwinTransformer) and point cloud transformers (e.g., PointTransformer v3).
The authors cover regular grids, particles, transformers, but miss some related literature for multiscale structures on irregular mesh, see below section for suggested refs to add.

**Theoretical Claims:**

NA; no concern: The method builds on mature techniques (ball trees, attention mechanisms) and provides solid incremental improvements.

---

> ### Author Rebuttal · Authors · 2025-03-25
>
> We thank the reviewer for their encouraging feedback and hope to address their concerns with the rebuttal.
>
> ## Experimental Designs Or Analyses
> > Not all baselines are applied to all datasets. For example, PointTransformer v3 is not included in the fluid dynamics experiments, and EAGLE is not included in the cosmology experiments. Suggestion: Include a comprehensive table listing all models and baselines for each dataset, with explanations for missing entries (e.g., due to incompatibility or heavy modifications required to the source).
>
> In our experiments we did not aim at evaluating every baseline on every dataset. For each benchmark, except for MD (as it is not an established benchmark), we report results for baselines that have been previously evaluated on the benchmark. That being said, we do compare extensively against PointTransformer v3 when possible as it is the state-of-the-art model from computer vision and is very close to Erwin in spirit as it also suggest a way to linearize attention. Unfortunately, as the reviewer pointed out, we did not evaluated it on the fluid dynamics dataset as PTv3 is only implemented for 3D, while the benchmark is 2D. Doing so would require heavy modifications from us, particularly in serialization (https://github.com/Pointcept/PointTransformerV3/blob/main/serialization/z_order.py), which is the cornerstone of the approach. If there are specific experiments the reviewer would like us to run for the rebuttal, we are happy to do it.
>
> **Action taken**: In the final version, we will further explain our logic in baseline selection in the experimental section for each benchmark stressing out the comparison against PTv3. We also conducted additional experiments on ShapeNet-Car which now includes comparison to Transolver - state-of-the-art model for PDE modelling.
>
> ## Supplementary Material
> > Only two datasets (cosmology, molecular dynamics) are introduced in detail. The fluid dynamics and airflow pressure datasets are mentioned briefly. Suggestion: Provide a coherent and complete introduction to all datasets, organized in a structured layout. Ensure all components (e.g., dataset statistics, preprocessing steps) are included.
>
> **Action taken**: We appreciate reviewer's suggestion and will use increased space in the final version to describe datasets in detail, particularly statistics and preprocessing steps. For the ease of comprehension, for each dataset, we will compile a table which describes
> - dataset statistics
> - train/validation/test split
> - input type (mesh/point cloud), feature type
> - target type
> - preprocessing (normalization, connectivity type, etc.)
>
> If there are any other descriptors that reviewers would like to be included, we would be happy to do so.
>
> ## Essential References Not Discussed
> **Action taken**: We appreciate the reviewer's suggestions and will use the increased page count to expand the related works section and include multiscale frameworks.
>
> ## Conclusion
> We are thankful to the reviewer for the valuable suggestions that we believe will improve the presentation of our framework. If the are any questions, we would be happy to answer them.

---

> > ### Comment · Reviewer_CTuS · 2025-04-02
> >
> > Dear authors and reviewers,
> >
> > I have read all the reviews and author comments. I appreciate the authors willing to take actions to a) explain why selecting some baselines while not for the others b) improving reproductivity and c) adding missing related works.
> >
> > However, as a) is my major concern (also some other reviewer's), and currently the authors did not provide a throughout explanation (they promise to do so in the final version; maybe a better idea is preparing some intermediate markdown tables, or figures in anounymous repo to explain), I believe my current rating is rather fair.
> >
> > I can keep an eye on if any more updates are provided.

---

> > > ### Author Response · Authors · 2025-04-02
> > >
> > > We appreciate reviewer's response to our rebuttal and would like to address concern a) explicitly with a comprehensive table as the reviewer suggested:
> > >
> > > | Model Type | Model | Cosmology¹ | MD | EAGLE¹ | ShapeNet² |
> > > |------------|-------|-----------|-----|-------|----------|
> > > | **Message-Passing Based** | MPNN | ✓ | ✓ | ✗ | ✗ |
> > > |  | SEGNN | ✓ | ✗ | ✗ | ✗ |
> > > |  | NequIP | ✓ | ✗ | ✗ | ✗ |
> > > |  | MGN | ✗ | ✗ | ✓ | ✗ |
> > > |  | GAT | ✗ | ✗ | ✓ | ✗ |
> > > | **Transformer-Based** | PointTransformer v3 | ✓ | ✓ | - | ✓ |
> > > |  | EAGLE | ✗ | ✗ | ✓ | ✗ |
> > > |  | OctFormer | ✗ | ✓ | ✗ | ✗ |
> > > |  | **Erwin (Ours)** | ✓ | ✓ | ✓ | ✓ |
> > > | **Other Hierarchical** | PointNet++ | ✗ | ✓ | ✗ | ✗ |
> > > | **Neural Operators** | U-Net | - | - | - | ✓ |
> > > |  | FNO | - | - | - | ✓ |
> > > |  | GINO | ✗ | ✗ | ✗ | ✓ |
> > > |  | UPT | ✗ | ✗ | ✓ | ✓ |
> > > |  | DRN | - | - | ✓ | - |
> > > |  | Transolver | ✗ | ✗ | ✗ | ✓ |
> > >
> > > **Legend:**
> > > - ✓: Model evaluated on this dataset
> > > - ✗: Model not evaluated on this dataset
> > > - -: Not applicable
> > >
> > > **Notes:**
> > > - ¹ An established benchmark with established baselines.
> > > - ² The benchmark is meant to compare neural operators against each other, hence only SOTA NOs are selected.
> > >
> > > **Rationale for Baseline Selection:**
> > >
> > > **Cosmology:**
> > > - We focus on comparing message-passing models (MPNN, SEGNN, NequIP) with sub-quadratic transformer models (PointTransformer v3, Erwin).
> > > - These baselines were selected to evaluate performance on large-scale systems with long-range interactions.
> > >
> > > **Molecular Dynamics:**
> > > - Comparison aimed at exploring the Pareto frontier (performance vs. runtime).
> > > - We selected hierarchical models that are scalable to large-scale systems, which is our primary focus.
> > >
> > > **Fluid Dynamics (EAGLE):**
> > > - We used the established benchmark with its native baselines (MGN, GAT, DRN, EAGLE).
> > > - PointTransformer v3 was not included as it's designed for 3D data, while this dataset is 2D.
> > >
> > > **ShapeNet-Car:**
> > > - Compared against existing neural operators and physics-based models (U-Net, FNO, GINO, UPT).
> > > - Added PointTransformer v3 as it's SOTA for large-scale point clouds.
> > > - Added Transolver as a SOTA model for large-scale physical simulations.
> > >
> > > We are grateful for the reviewer's feedback and we hope that this response clarifies the baseline selection.

---

### Official Review · Reviewer_qhYr · 2025-03-09

**Overall Recommendation:** 4

**Summary:**

The authors propose a tree-based hierarchical transformer model to capture large-scale and small-scale interactions in exhaustively large datasets. The motivation is that traditional transformers are not scalable and efficient, so the authors propose a hierarchical model to allow course granting to capture large scale interactions that scales ~linearly with the data size. They test the proposed model on three empirical data sets.

**Claims And Evidence:**

The claims around computational efficiency, linear scalability, and performance are evaluated empirically and backed by different experiments.

**Essential References Not Discussed:**

N/A

**Experimental Designs Or Analyses:**

The cosmology experiment has some logical gaps. This paper takes the location of galaxies (halos) and wants to predict their velocity. Location and velocity, even though correlated, are two independent random variables. Their correlation is due to the initial condition, therefore the model is not generalizable to another initial condition or universe with different properties. I am not sure what the motivation is behind setting up the problem in this way.

My intuition is that writing the loss function as the MSE of log Y for the turbulent fluid dynamics problem is more appropriate. The fractional error is important, not the actual value. The actual value has a unit and two quantities with different units (strictly speaking) should not be added together. However, fractional error does not have a unit and the fractional error of two quantities can be added.

**Methods And Evaluation Criteria:**

The methods and evaluation criteria, including the benchmarks, are suitable for the problem and demonstrate the superiority of the proposed solution.

**Other Comments Or Suggestions:**

- "halos form local clusters through gravity while maintaining long-range correlations originated from interactions in the early universe before cosmic expansion" -> the universe has been expanding since the big bang. Also, the long range correlations are vanishing (unless a non-standard cosmology is assumed).

**Other Strengths And Weaknesses:**

The main target audience is people who use AI for science applications, and I feel this ICML is an appropriate venue for this kind of publication.

**Questions For Authors:**

- "The input is a point cloud $X \in \mathcal{R}^{5000\times3}$" -> I assume 3, the the 3-spacial dimensions. Can the authors elaborate on what time the positions and velocity vectors are measured?

- "The total dataset includes 1184 simulations with 990 time steps per simulation. The dataset is split with 80% for training and 10% each for validation and testing." -> Can you describe how the training, validation, and test are selected?

- I assume the proposed solution has application mainly to problems where data are in d-dimensional spatial domain. Are there any other applications that the author can think of where the unit balls are not defined in the spatial domain?

**Relation To Broader Scientific Literature:**

The proposed solution si appropriate for the target problems. The authors consider a set of scientific problems in which the traditional transformer-based models are not scalable and limited. Therefore, they propose a hierarchical solution that is suitable for the class of problems studied in this work and can be readily applied to those problems.

**Theoretical Claims:**

N/A. No theoretical claims are made in this paper.

---

> ### Author Rebuttal · Authors · 2025-03-25
>
> We thank the reviewer for their positive feedback. We shall address their questions below. When discussing the questions regarding the cosmology dataset, we will refer to the original dataset paper by Balla et al.
>
> ## Questions
>
> > "The input is a point cloud $X \in \mathbb{R}^{5000 \times 3}$" -> I assume 3, the the 3-spacial dimensions. Can the authors elaborate on what time the positions and velocity vectors are measured?
>
> Yes, that is correct - 3 stands for spatial dimensions. Regarding the measurement time, according to A.1. Details of simulation of Balla et al., it is the last step of a Quijote simulation (*present time*): "Each simulation models the evolution of the large-scale structure of the Universe by following the dynamics of 5123 cold dark matter particles in a cubic comoving volume of side ∼ 1 Gigaparsec from redshift z = 127 to z = 0 (present time)." The simulations are defined in Villaescusa-Navarro et al.
>
> > "The total dataset includes 1184 simulations with 990 time steps per simulation. The dataset is split with 80% for training and 10% each for validation and testing." -> Can you describe how the training, validation, and test are selected?
>
> Since we use the benchmark, we use the predefined validation and test partitions as defined in Balla et al. (See https://zenodo.org/records/11479419 and https://github.com/smsharma/eqnn-jax/blob/main/benchmarks/galaxies/dataset.py for details). Training samples are taken randomly from the total number of 11,200.
>
> > I assume the proposed solution has application mainly to problems where data are in d-dimensional spatial domain. Are there any other applications that the author can think of where the unit balls are not defined in the spatial domain?
>
> 1) One application that comes to mind is the relativistic case where particles live in spacetime, and one would need to adjust the algorithm to account for the pseudo-Euclidean structure.
> 2) Related to spacetime, if one axis represents time, unit balls will not make much sense. In this scenario, however, one can simply build a ball tree based on the spatial coordinates and omit time during the construction. After all, a ball tree can be seen just as a means of organizing computation to linearize attention.
>
> ## Other comments
>
> > "halos form local clusters through gravity while maintaining long-range correlations originated from interactions in the early universe before cosmic expansion" -> the universe has been expanding since the big bang. Also, the long range correlations are vanishing (unless a non-standard cosmology is assumed).
>
> Regarding the universe expansion, we thank the reviewer for their careful reading and for identifying this textual error. Indeed, what we meant to say was that the structures were initially in causal contact. We will remove "before cosmic expansion" in the revised manuscript. Regarding long-range correlations, although vanishing, they are still present according to the benchmark description (see Fig. 1 and "Introduction, Information across scales" in Balla et al.).
>
> ## Experimental Designs Or Analyses
>
> We appreciate the reviewer's critical assessment of the benchmarks used in our evaluation. We note that both cosmology and EAGLE datasets are benchmarks with established tasks/preprocessing/partitioning/losses. That is the reason we predict velocity from positions - it is the original task of the cosmology benchmark (see Balla et al., Section 3.3, Node-level prediction). For the same reason, we stick to the losses as proposed in the EAGLE benchmark. Overall, we used the original settings defined in the benchmarks for consistent comparison with other baselines validated on each benchmark.
>
> ## Conclusion
>
> We thank the reviewer for the valuable feedback and are hopeful that the reviewer will continue to support the paper. If the are any questions, we would be happy to answer them.
>
> ## References
>
> - Balla, J., Mishra-Sharma, S., Cuesta-Lázaro, C., Jaakkola, T., & Smidt, T.E. (2024). A Cosmic-Scale Benchmark for Symmetry-Preserving Data Processing. Learning on Graphs Conference, 2024.
>
> - Villaescusa-Navarro, F., Hahn, C., Massara, E., Banerjee, A., Delgado, A.M., Ramanah, D.K., Charnock, T., Giusarma, E., Li, Y., Allys, E., Brochard, A., Chiang, C., He, S., Pisani, A., Obuljen, A., Feng, Y., Castorina, E., Contardo, G., Kreisch, C.D., Nicola, A., Scoccimarro, R., Verde, L., Viel, M., Ho, S., Mallat, S., Wandelt, B.D., & Spergel, D.N. (2019). The Quijote Simulations. The Astrophysical Journal Supplement Series, 250.

---

### Official Review · Reviewer_eeiZ · 2025-03-11

**Overall Recommendation:** 3

**Summary:**

The paper introduces Erwin, a hierarchical transformer model designed for large-scale physical systems. The model employs a ball tree partitioning strategy to group nodes, enabling linear-time attention by processing local neighborhoods in parallel. This hierarchical approach allows progressive coarsening and refinement, capturing both fine-grained local details and global interactions. The paper demonstrates Erwin’s effectiveness across multiple domains, including cosmology, molecular dynamics, and particle fluid dynamics.

**Claims And Evidence:**

Most claims are supported.

**Essential References Not Discussed:**

The paper includes most of the relevant prior work and provides thorough comparisons. However, since one of Erwin’s key contributions is hierarchical modeling, it might be beneficial to discuss and compare with existing hierarchical models in physical systems, such as BSMS-GNN [1] and HCMT [2]. These models incorporate multi-scale representations in physics simulations and could provide additional context for understanding Erwin’s approach.

[1] Efficient Learning of Mesh-Based Physical Simulation with BSMS-GNN.

[2] Learning Flexible Body Collision Dynamics with Hierarchical Contact Mesh Transformer.

**Experimental Designs Or Analyses:**

Yes.

While the paper provides training information for Erwin, it lacks precise training details of compared methods.

**Methods And Evaluation Criteria:**

Yes.

**Other Comments Or Suggestions:**

Where is the anonymous link?

**Other Strengths And Weaknesses:**

**Strengths**
- The paper presents a creative combination of ball tree partitioning and transformer-based attention, leading to an efficient and expressive hierarchical model for large-scale physical systems.
- By leveraging ball tree partitioning, the model reduces computation cost, which is important for large-scale physics systems.

**Weakness**
- The paper discusses OctFormer, another tree-based hierarchical attention model, but does not include a direct comparison. Since both methods aim to improve efficiency through hierarchical partitioning, an empirical evaluation would better highlight Erwin’s advantages.
- While the paper compares Erwin to UPT and EAGLE, which use two-level hierarchical pooling, it does not consider more flexible multi-level approaches like BSMS-GNN[1] and HCMT[2], which incorporate multi-scale message passing.
- Erwin performs worse than some baselines when trained on small datasets. This suggests that the model may rely more heavily on large-scale data to learn effective representations, whereas certain baselines generalize better with limited training samples.
- Table 2 shows that most improvements come from MPNN in the encoder, while Erwin Blocks (ball attention) contribute less. This raises the question of whether hierarchical attention is essential, or if similar gains could be achieved with a strong graph-based encoder and standard attention.
- The paper directly adopts baseline results from EAGLE for turbulent flow experiments, but it is unclear whether Erwin’s model depth is aligned with the baselines. Additionally, the appendix lacks details on Erwin’s architecture for this experiment, making it difficult to assess the fairness of the comparison.
- Erwin’s MPNN and the compared MPNN baseline rely on k-nearest neighbors, while a more general and physically meaningful approach is ball neighborhoods, as used in MGN paper.
- Some important methodological and experimental details are missing (see questions), which significantly affect my recommendation.

**Questions For Authors:**

1. In Figure 9, the ball tree partitions appear to vary significantly across different datasets. Did you analyze how the choice of tree depth or splitting strategy affects performance? For example, in molecular dynamics, does the partitioning align well with physical structures like polymer chains, or are there cases where the tree creates unnatural groupings?
2. The partitions in ShapeNet and EAGLE seem mostly aligned with coordinate axes. Given this, how does ball tree partitioning compare to KD-tree partitioning for these datasets? Would KD-trees provide similar efficiency, and what advantages does the ball tree offer in these cases?
3. Erwin pads the ball tree to form a perfect binary tree, but the paper does not specify how virtual nodes are positioned or initialized. Are they assigned based on nearby real nodes, or are they purely placeholders? Could learnable pooling or adaptive padding reduce computational overhead?
4. Table 1 ablates ball size for cosmology dataset, but how was ball size chosen for each datasets?
5. How many layers does the MPNN in the embedding stage have?
6. The paper introduces cross-ball connections by rotating the point cloud before constructing a second ball tree. What specific rotation matrix is used? Is it a fixed transformation, randomly sampled, or learned? How does the choice of rotation affect performance?
7. The results suggest that MPNN performance plateaus as training data increases, but my experience, especially MGN models, is that they usually continue improving. Can you give me a suitable explanation?
8. Did you ensure that all baseline models have comparable depth and capacity with Erwin?

A clear response to these questions would help clarify my key concerns and could positively influence my evaluation.

**Relation To Broader Scientific Literature:**

Erwin builds on transformer-based approaches, but introduces hierarchical ball tree partitioning to improve efficiency.

**Theoretical Claims:**

Yes.

---

> ### Author Rebuttal · Authors · 2025-03-26
>
> We thank the reviewer for the actionable feedback. We address their concerns and questions in the following.
>
> ## Questions For Authors
> ### Q1
>
> That is an insightful question. In general, ball trees are geometry-adaptive but do not guarantee that balls will cover points in a perfectly natural way and certain irregularities can indeed appear in practice. To address it, we introduce the distance-based attention bias (Eq. 10), which penalizes the strength of interaction between distant tokens. It provides a soft mechanism for creating more “natural” partitions. In our opinion, such partitioning irregularities are a byproduct of *any* attempt to impose a rigid structure onto irregular data, such as converting a point cloud into a sequence (e.g. space-filling curves in PTv3 produce discontinuities).
>
> Regarding tree depth, we clarify that it is fixed, as the tree is always constructed until leaf size 2 and is not a variable design parameter. For the splitting strategy, we employ Algorithm 1, selected primarily for its speed (also used by scikit-learn).
>
> ### Q2
>
> For both these datasets ball tree partitioning aligns very well with KD-tree partitioning and KD-tree would be a viable alternative. However, one advantage of ball trees is that they offer a rotation invariant partitioning which might be critical for certain physical applications.
>
> ### Q3
>
> We are thankful to the reviewer for bringing our attention to it and apologize for overlooking it in the original submission. We will include details in the final version. Tree construction creates leaf nodes with either 1 point (incomplete) or 2 points (complete). For single-point leaves, we duplicate the point as padding, creating virtual nodes that are copies of original points. This simplifies implementation, allowing pooling to work without edge cases and minimal computational overhead. Speaking of learnable pooling, this is one of the directions for future work as it would make Erwin applicable to gigantic-scale industrial applications and we aim to explore it.
>
> ### Q4
>
> Ball size is chosen according to computational constraints, i.e. available GPU memory. In our experiments, it was a hyperparameter which we tuned for the best performance, choosing between 128 and 256. Generally, it is a trade-off as 256 provides better coverage but 128 allows for using larger batch size.
>
> ### Q5
>
> We specify the number of  message-passing steps in Tables 6 and 7 for ShapeNet ($1$ step) and MD ($2$ steps). We will also add the detail for EAGLE and cosmology, where the number of steps was $3$.
>
> ### Q6
>
> For 2D cases, we specify degrees of rotation, and for 3D, we specify Euler angles. In all experiments we used fixed value of 45 which we empirically found to generate sufficiently different partitions. The difference between partitionings influences the performance significantly, as it is the mechanism that allows points within different original balls to interact. Furthermore, we avoided using small angles, but there was no difference for sufficiently large angles.
>
> ### Q7
>
> The reviewer refers to the cosmology experiment where the performance of MP models plateaus with increasing number training points. This is consistent with results of the original benchmark paper (Balla et al., Fig. 3). In the task, each data point is a point cloud with 5000 points. Furthermore, even for small training sizes, there is enough signal to capture local interactions for message-passing models. We note, however, that there are also long-range dependencies in data (Balla et al., Fig. 1), which MP models are not able to capture. Furthermore, regardless of how much data is given to them, they are not able to get that part of the signal due to short-range receptive field, which is different for Erwin or PointTransformer v3 that both have large receptive fields.
>
> ### Q8
>
> We did control explicitly in *all* the tasks that both Erwin and state-of-the-art PT v3 have the same depth and similar number of parameters as those models are comparable in their architecture.
> - cosmology: we use the same hyperparameters for each model as in the original paper (Balla et al.), as those are the best performing models according to the hyperparameter tuning from the authors (see Appendix C, Balla et al.).
> - EAGLE: we take results from the original paper where authors do hyperparameter sweep and thus report best-performing configurations. Parameter-wise, Erwin has 18M params and the closest competitor, EAGLE, has 10M params.
> - ShapeNet, all models have similar parameter count - between 4M and 5M parameters.
>
> ## Conclusion
> Unfortunately, we could only use 5k characters for the rebuttal and do not have space to address weaknesses one-by-one despite having prepared answers. For additional experiments that address W1 and W4, we kindly refer to another rebuttal: https://openreview.net/forum?id=MrphqqwnKv&noteId=BfX9JC9Pxg. We would be happy to engage in further discussion if the reviewer would like us to clarify further.

---

> > ### Comment · Reviewer_eeiZ · 2025-04-02
> >
> > I have read all the reviews and author comments. Thank you for your response and for providing the missing details, which address most of my concerns. I would appreciate it if these additional discussions and details could be incorporated into the revised manuscript, as they would greatly help in understanding the method. If you commit to doing so, I will consider increasing my rating.
> >
> > Regarding Q6, there is still a small issue: Could you clarify what "sufficiently different" means in this context? How do the partitions differ, and is it possible to quantify this difference?

---

> > > ### Author Response · Authors · 2025-04-02
> > >
> > > We promise to include the additional details/results and would like to emphasize our commitment to doing it.
> > >
> > > *Regarding Q6*, that is perhaps easier to explain on the examples:
> > > https://github.com/erwin-transformer/erwin/blob/main/misc/ball_tree_with_rotations.png
> > >
> > > Here, you can see that partitions are different in a sense that the **set difference** between two partitions that cover the same original point is high. Take a look, for example, on the yellow partitions. They cover different sets of points in both original and rotated configurations, however, there is still a substantial intersection. This allows points that were originally in the right upper corner to technically interact with points that were in the left upper corner (what we call information leakage).
> > >
> > > Coming back to your question, we believe that we can robustly quantify the difference by measuring the set difference on the highest but one level of the tree (when there are only two partitions).

---

### Official Review · Reviewer_dSeu · 2025-03-17

**Overall Recommendation:** 3

**Summary:**

The paper introduces a transformer-based approach for processing point cloud data. It utilizes ball-tree-based structures, enabling the attention-based framework to operate in linear time rather than quadratic time. This also allows the model to capture interactions at both fine and coarse scales.

The proposed method is evaluated on three problems: capturing long-range interactions (cosmology); computational efficiency (molecular dynamics); model expressivity on large-scale multi-scale phenomena (turbulent fluid dynamics), demonstrating strong performance in both computational efficiency and prediction accuracy.


update after rebuttal
Thank the authors for the responses. I decide to keep my original score.

**Claims And Evidence:**

The claims made in the submission is supported by clear and convincing evidence.

**Essential References Not Discussed:**

no.

**Experimental Designs Or Analyses:**

The experimental designs make sense.

**Methods And Evaluation Criteria:**

The proposed methods and evaluation criteria make sense.

**Other Comments Or Suggestions:**

n/a

**Other Strengths And Weaknesses:**

The proposed ball-tree-based attention seems novel to me.
Besides the algorithm, the paper also discusses the implementation details of the ball-tree attention, such as, the representation of the tree which makes the tensor rearranging on the nodes more efficient.

**Questions For Authors:**

- The paper mentions "The code is available at anonymized link". But there seems to be no link attached at the "anonymized link".
- During the simulation process, the positions of particles will change. Does the proposed method re-build the tree after each step?

**Relation To Broader Scientific Literature:**

The paper discussed the existing works of sub-quadratic attention. Compared to those works, the paper uses ball-tree structures to capture the hierarchical structures of point cloud data.

**Theoretical Claims:**

n/a

---

> ### Author Rebuttal · Authors · 2025-03-25
>
> We thank the reviewer for their encouraging feedback. We address their questions below.
>
> ## Questions For Authors
> > The paper mentions "The code is available at anonymized link". But there seems to be no link attached at the "anonymized link".
>
> We apologize for the inconvenience. An anonymized version of the repo can be found here: https://github.com/erwin-transformer/erwin.
>
> > During the simulation process, the positions of particles will change. Does the proposed method re-build the tree after each step?
>
> Rebuilding the tree depends entirely on the properties of the modelled system, particularly, how quickly the arrangement of particles changes:
> - The mesh is *static*: the tree should only be computed once during preprocessing.
> - The mesh changes *slowly*: the tree should be rebuilt each N step.
> - The mesh is dynamic and changes *quickly*: ideally, the tree should be rebuilt at every step of a simulation.
>
> During the code implementation, we assumed that we would need to build the tree as fast as possible. Hence, we implemented ball tree construction in C++ & OpenMP and according to our benchmarks, it consistently takes less than 10% of the forward pass. For details, see https://github.com/erwin-transformer/erwin?tab=readme-ov-file#benchmark.
>
> ## Additional experiments
>
> We also conducted additional experiments to further highlight the strength of our method.
> ### OctFormer on the molecular dynamics task
>
> We will include OctFormer results for the molecular dynamics task in the final manuscript. We have evaluated OctFormer while maintaining similar parameter count, depth, and partitioning size (128) as Erwin for each model size.
> |  Model || model size ||
> |--------------------------------|------------|--------------|-------------|
> |                               | small (4M) | medium (19M) | large (43M) |
> | OctFormer                      | 0.725      | 0.715        | 0.712       |
> | Erwin                          | **0.712**      | **0.693**        | **0.691**   |
>
> ### Ablation study on ShapeNet-Car dataset
> We conducted an ablation study to gather additional information concerning the role of the MPNN embedding and Ball Attention in Erwin's performance. Here, Erwin has a small size (1.5M parameters) and a fixed width (we do not use any upsampling/downsampling to achieve the best performance)
>
> | | Test MSE |
> |-------------|--------|
> | w/o         | 30.39  |
> | + MPNN      | 30.49  |
> | + RPE       | 30.02  |
> | + Rotating trees | **16.58** |
> | Transolver (3.9M) | 19.88 |
>
> Notice that in this experiment, including the MPNN leads to deteriorating performance, while rotating trees essentially allow Erwin to achieve state-of-the-art performance. The reason is that the ball attention is able to process large-scale data (3.6K points) at full fidelity, and rotating trees allow for information to propagate from one ball to another, essentially making the receptive field encompass the full car.
>
> ## Conclusion
>
> We are grateful for the reviewer's feedback. Should you have any questions or concerns, we are always ready to discuss them further.

---

### Decision · Program_Chairs · 2025-05-01

**Decision:**

Accept (poster)

**Comment:**

This paper presents Erwin, a tree-based hierarchical transformer tailored for large-scale physical systems defined on irregular grids. By leveraging ball tree partitioning and cross-ball interactions, Erwin achieves linear-time attention, capturing both local and global interactions effectively. The model is evaluated across diverse scientific domains, including cosmology, molecular dynamics, and turbulent fluid simulations, and demonstrates improvements in both accuracy and computational efficiency.

The authors spent quite some time on the rebuttal dispersing some/most(?) of the initial criticism which lead to increased scores and the general sentiment is in favor of the paper (averaging around weak accept) and so am I. I recommend weak acceptance; a bit of rework could have improved the submission further and I sincerely hope that the authors are doing so in their camera-ready or the revision.